# Snord67 promotes breast cancer metastasis by guiding U6 modification and modulating the splicing landscape

Yvonne L. Chao[1,2,3,4,5,16], Katherine I. Zhou[1,6,7,16], Kwame K. Forbes[7,8], Alessandro Porrello[1,7], Gabrielle M. Gentile[9,10], Yinzhou Zhu[11], Aaron C. Chack[1,7,10], Dixcy J. S. John Mary[3], Haizhou Liu[3], Eric Cockman[11], Lincy Edatt[1,7], Grant A. Goda[7,12,13], Justin J. Zhao[11], Hala Abou Assi[11], Hannah J. Wiedner[9,10], Yihsuan Tsai[1], Lily Wilkinson[1], Amanda E. D. Van Swearingen[1], Lisa A. Carey[1,2], Jimena Giudice[7,9,10,14], Daniel Dominguez[1,7,8,13,15] ✉, Christopher L. Holley[11] ✉ & Chad V. Pecot[1,2,7] ✉

Previously considered "housekeeping" genes, small nucleolar RNAs (snoRNAs) are increasingly understood to have wide-ranging functions in cancer, yet their role in metastasis has been less well studied. Here, we identify the snoRNA Snord67 as a regulator of lymph node (LN) metastasis in breast cancer. Snord67 expression is enriched in LN metastases in an immune-competent mouse model of female breast cancer. In an orthotopic breast cancer model, loss of Snord67 decreases LN metastasis. In a model of lymphatic metastasis, Snord67 loss decreases LN tumor growth and distant metastases. In breast cancer cell lines, Snord67 knockout results in loss of targeted 2′-O-methylation on U6 small nuclear RNA, as well as widespread changes in splicing. Together, these results demonstrate that Snord67 regulates splicing and promotes the growth of LN metastases and subsequent spread to distant metastases. SnoRNA-guided modifications of the spliceosome and regulation of splicing may represent a potentially targetable pathway in cancer.

Distant metastases are the primary cause of cancer-related mortality, and lymph node (LN) metastases correlate with both increased risk of distant metastasis as well as poor prognosis[1]. However, the lack of a clear survival benefit from the surgical resection of axillary LNs (AxLNs) in breast cancer suggests that while LN metastases are clinically important, they have a complex relationship with distant metastases, and the factors that determine when and how LN metastases give rise to distant metastases remain poorly understood[2]. Although LN metastases have historically been considered a surrogate for the ability of cancer cells to metastasize via hematogenous dissemination, several studies have demonstrated that LN metastases can directly give rise to distant metastases via lymphatic spread[3–5]. Moreover, the LN harbors a unique microenvironment to which cancer cells must adapt. For

example, the LN is a fatty acid-rich microenvironment, and cancer cells adapt by upregulating lipid metabolism pathways[6]. In melanoma, LN colonization has been shown to result in epigenetic rewiring of both tumor and immune cells, which in turn promotes distant metastases[7]. However, the mechanisms governing the survival of cancer cells in LNs and the spread of cancer cells to distant organs from LNs remain poorly understood.

To investigate the mechanisms that drive LN metastasis, we previously developed a microsurgical murine model using 4T1 breast cancer cells, which form tumors that histopathologically resemble triple-negative breast cancer but have a gene expression profile more consistent with the luminal molecular subtype[8,9]. Using this model, we demonstrated that micro-injected AxLN tumors can spontaneously

establish distant lung metastases and that metastasis via the lymphatic route is more efficient than hematogenous dissemination[8]. To further explore the mechanisms by which LN tumors grow and give rise to distant metastases, we evaluated protein-coding genes that were differentially expressed in micro-injected AxLNs[8]. However, besides protein-coding genes, non-coding RNAs (ncRNAs) also function in the dynamic regulation of gene expression in cancer[10–14]. Notably, microRNAs (miRNAs) and long ncRNAs have well-established roles in metastatic biology through the pre- and post-transcriptional regulation of cell fate[15,16]. In contrast, while some small nucleolar RNAs (snoRNAs) have been identified as either oncogenic or tumor suppressive, the role of snoRNAs in metastasis remains largely unexplored[17–20]. SnoRNAs are small ncRNAs less than 300 nucleotides in length that primarily guide the site-specific modification of target RNAs via sequence complementarity at a region called the antisense element (Supplemental Fig. 1A). The two subtypes of snoRNAs, box C/D and box H/ACA, guide either 2′-O-methylation (box C/D) or pseudouridylation (box H/ACA) of their target RNAs. The canonical targets of snoRNAs are ribosomal RNAs (rRNAs) and small nuclear RNAs (snRNAs), but snoRNA-guided modification of mRNA targets has also been demonstrated[21–25]. In addition, some snoRNAs have been shown to function through non-canonical mechanisms that do not involve 2′-O-methylation or pseudouridylation[13].

In this study, we use our microsurgical murine model of LN metastasis to identify ncRNAs that are differentially expressed in LN metastases compared to primary tumors and distant metastases. We find that snoRNAs are frequently upregulated in LN tumors and identify the box C/D snoRNA Snord67 as one of the most upregulated snoRNAs. We further demonstrate that loss of Snord67 in breast cancer cell lines leads to decreased colony formation and spheroid area in vitro, as well as decreased lymph node tumor growth and distant metastases in murine models. Finally, we investigate possible mechanisms by which Snord67 might promote lymphatic metastasis, showing that Snord67 is required for site-specific 2′-O-methylation of U6 snRNA and that decreased Snord67 expression is associated with widespread changes in splicing patterns in murine and human breast cancer cell lines. We further validate two Snord67-dependent differential alternative splicing events that also correlate with Snord67 expression in paired primary breast and lymph node tumors from patients with breast cancer. Together, our results reveal that Snord67 promotes the lymphatic dissemination of breast cancer and suggest the Snord67-mediated regulation of splicing as a potential mechanism.

## Results

### Many snoRNAs are upregulated in lymph node metastases
Using the 4T1 model, we generated subclones of cancer cells from mammary fat pad (MFP) tumors, microsurgically-injected AxLN tumors (Supplemental Fig. 1B), and spontaneous lung metastases derived from AxLN tumors (AxLN-LuM) (Supplemental Fig. 1C). We performed microarray profiling of these subclones to identify ncRNAs that were differentially expressed in AxLN tumors compared to MFP tumors and AxLN-LuM (Fig. 1A, Supplemental Fig. 1D, E, and Supplementary Data 1, 2). Among 10,043 probe sets corresponding to ncRNAs and poorly characterized RNAs on the microarray, only 30 demonstrated differential expression in AxLN subclones compared to MFP and AxLN-LuM (Supplemental Table 1). In this subset of differentially expressed ncRNAs, 67% of the probe sets corresponded to snoRNAs (20 out of 30). This was a striking finding since snoRNAs comprised only 12% of all RNAs profiled in this study (1245 out of 10,043; Fig. 1B and Supplemental Table 1). Thus, there was a more than five-fold, statistically significant enrichment of snoRNAs among the differentially expressed ncRNAs ($p = 5.6 \times 10^{-12}$ by the hypergeometric test). On the other hand, most snoRNAs (98%) profiled on the microarray were not differentially expressed in AxLN tumors relative to MFP tumors and AxLN-LuM. Thus, a small subset of snoRNAs is responsible for a

large fraction of the changes observed in the non-coding transcriptome of AxLN tumors.

The top six box C/D snoRNAs identified in the profiling analyses were validated using reverse transcription and quantitative polymerase chain reaction (RT–qPCR), which confirmed increased expression of these snoRNAs in AxLN subclones compared to MFP tumors and AxLN-LuM, with the most significant changes observed in Snord67 and Snord111 (Fig. 1C). To determine whether Snord67 and Snord111 expression also increased in de novo LN metastases, we measured expression of these snoRNAs by RT–qPCR in matched pairs of MFP tumor subclones and de novo LN metastasis subclones. For the majority of the matched pairs, expression of both Snord67 and Snord111 was increased in the de novo LN metastasis subclones compared to the MFP subclones (Fig. 1D). Together, these results show that the box C/D snoRNAs Snord67 and Snord111 are consistently upregulated in both microsurgically-injected LN tumors and de novo LN metastases.

### Loss of Snord67 decreases proliferation and tumorigenesis
Since Snord67 and Snord111 are specifically upregulated in LN metastases, we hypothesized that these snoRNAs may function in the lymphatic route of metastasis. To evaluate the function of these snoRNAs in 4T1 breast cancer cells, we generated Snord67 and Snord111 knockout clones using the CRISPR/Cas9 system. Due to concerns that a single genomic deletion would not be sufficient to knock down expression of a snoRNA[26], we employed a double-nicking strategy using paired guide RNAs to reduce off-target mutagenesis (Fig. 1E)[27]. Two single-cell knockout clones were isolated for each of the two snoRNAs, Snord67 and Snord111, and genomic deletions at the predicted sites were confirmed by DNA sequencing. The double-nicking strategy resulted in clones with either two separate deletions or a larger deletion spanning the target sites. Loss of snoRNA expression in Snord67 and Snord111 knockout clones was verified by RT–qPCR (Supplemental Fig. 2A). SnoRNAs are typically encoded within introns of host genes, and regulation of snoRNA expression can be dependent or independent of host gene transcription levels[28,29]. Therefore, we verified that the expression of host genes *CKAP5* (Snord67) and *SF3B3* (Snord111) were not affected in the knockout clones (Supplemental Fig. 2B, C). CKAP5 protein expression and CKAP5 mRNA splicing in the region flanking Snord67 were also unaffected by Snord67 knockout (Supplemental Fig. 2D, E). We then evaluated the functional significance of loss of Snord67 or Snord111 in breast cancer cells. Compared to 4T1 wild-type (WT) cells, loss of Snord67 expression resulted in decreased colony formation and proliferation, while loss of Snord111 had no effect on either phenotype (Fig. 1F–H and Supplemental Fig. 3A–C). Migration was not affected by loss of either Snord67 or Snord111 (Supplemental Fig. 3D). Since Snord67 knockout cells exhibited decreased in vitro proliferation and colony formation, we focused on Snord67 in subsequent experiments.

Having demonstrated that Snord67 is necessary for in vitro proliferation and colony formation in 4T1 cells, we next investigated whether Snord67 expression is sufficient to rescue these in vitro phenotypes. Transfection with in vitro transcribed Snord67 led to only transient rescue (around 24 h) of Snord67 levels in 4T1 cells, so instead, we stably transfected one of the 4T1 Snord67 knockout clones (Snord67KO-2) with a Snord67 expression construct (OE). The putative function of Snord67 is to guide 2′-O-methylation at the C60 position in U6 snRNA, which is a core component of the spliceosome. To investigate whether Snord67-guided 2′-O-methylation of U6 C60 is important for its effect on in vitro phenotypes, we generated a mutant Snord67 construct with a point mutation (G97C) in its antisense element, which is expected to abolish Snord67-guided 2′-O-methylation of U6 at C60 (Fig. 2A)[30]. Snord67KO-2 cells were then transfected with this mutant Snord67 expression construct (mutOE). Snord67 expression was restored to approximately physiologic levels in Snord67KO-2

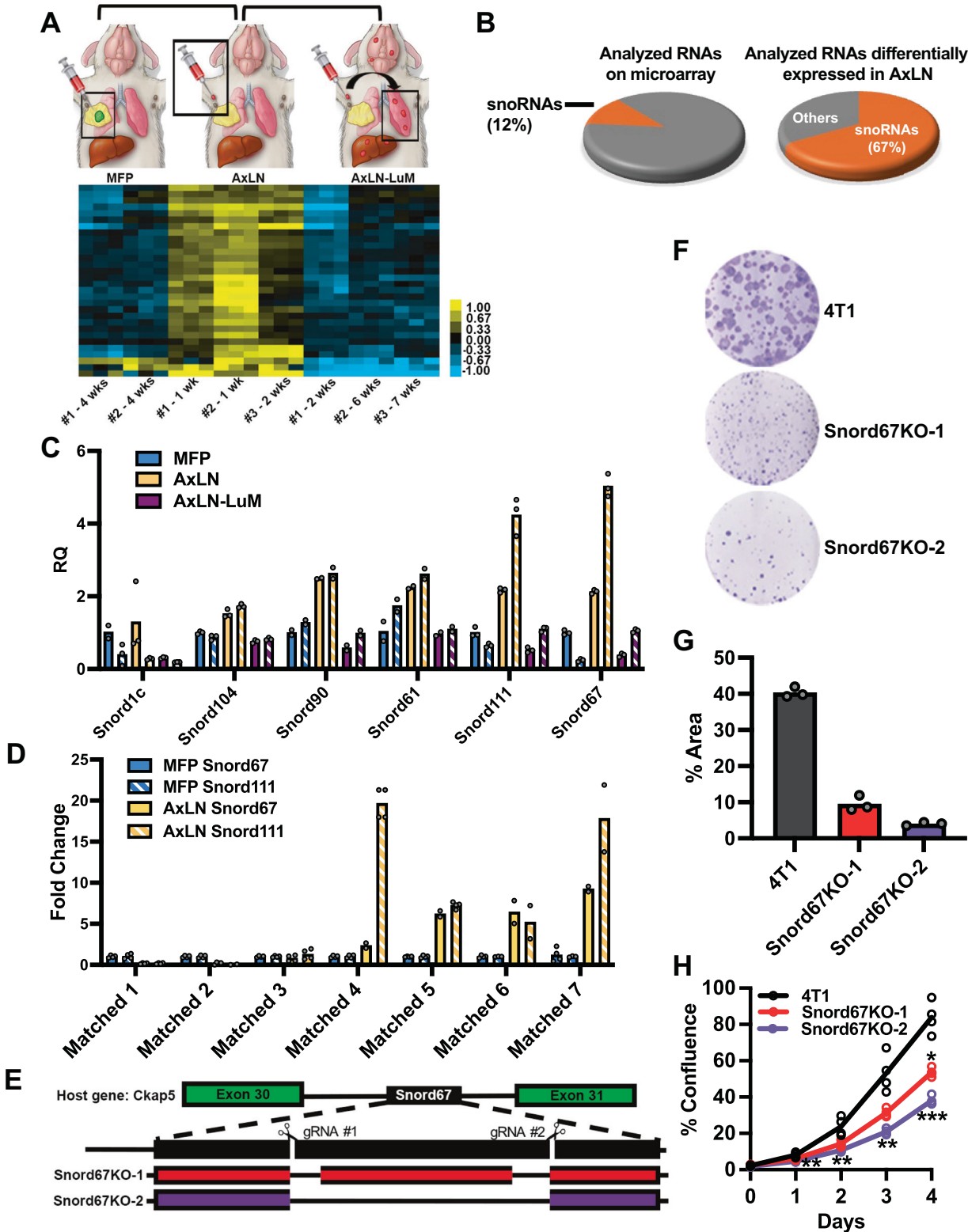

cells stably transfected with the wild-type and mutant Snord67 constructs (OE and mutOE) but not in cells transfected with empty vector (EV) (Fig. 2B). We then measured 2′-O-methylation at C60 in U6 snRNA using the Nm-VAQ (2′-O-methylation Validation and Absolute Quantification) method, in which site-specific 2′-O-methylation is measured based on the fraction of transcripts that are protected from RNase H cleavage (Supplemental Fig. 4A)[31]. Nm-VAQ on mixtures of RNA from

WT and Snord67 knockout cells showed that percent U6 C60 methylation increased linearly with increasing fraction of WT RNA, indicating that Nm-VAQ provides a quantitative measurement of percent U6 C60 methylation (Supplemental Fig. 4B). In 4T1 cells, loss of Snord67 resulted in a significant decrease in U6 C60 2′-O-methylation without affecting U6 snRNA abundance (Fig. 2C and Supplemental Fig. 4C). The wild-type Snord67 expression construct (OE) led to partial rescue of

**Fig. 1 | Identification and validation of snoRNAs that are upregulated in LN tumors. A** Composite microarray showing ncRNAs and poorly characterized RNAs that were differentially expressed in axillary lymph node (AxLN) tumors (*n* = 3) compared to mammary fat pad (MFP) tumors (*n* = 2) and lung metastases derived from AxLN tumors (AxLN-LuM) (*n* = 3). Tumors were harvested at indicated time-points, expanded ex vivo to generate subclones, and run in 3 biological replicates. Expression values were plotted on the heat map using colors (yellow-blue) assigned by the TreeView plotting software, with the contrast set to 1 (see the color bar). Graphics © 2020 The University of Texas M.D. Anderson Cancer Center. **B** Pie charts showing the proportion of snoRNAs (orange) among ncRNAs and poorly characterized RNAs on the microarray (*left*) and among analyzed RNAs differentially expressed in AxLN tumors versus MFP tumors and AxLN-LuM (*right*). **C** Relative quantification (RQ) of snoRNAs identified on the microarray in AxLN and AxLN-LuM subclones versus MFP by RT–qPCR. *n* = 2 subclones per tumor site, 3 technical replicates. **D** RQ of Snord67 and Snord111 in de novo LN metastases versus

matched MFP tumors by RT–qPCR. *n* = 7 pairs of subclones, *n* = 4 technical replicates (except n = 3 technical replicates for Matched 6–7 MFP Snord111, n = 2 technical replicates for Matched 4–7 AxLN Snord67 and Matched 2, 6, and 7 AxLN Snord111). **E** Schematic of the CRISPR/Cas9 double-nicking strategy for snoRNA knockout and the deletions observed in the Snord67 knockout (Snord67KO) clones. **F, G** Colony formation assay of 4T1 WT cells (4T1) and Snord67KO cells quantified by area of well coverage, plated in 3 replicates, repeated independently at least 4 times with similar results. **H** Proliferation of 4T1 WT and Snord67KO cells, quantified as percent confluence in time-lapse images captured using IncuCyte over 5 days (line = mean, *n* = 4 biological replicates). Adjusted \**p* < 0.01, \*\**p* < 0.001, \*\*\**p* < 0.0001. Statistical analysis was by unpaired t-tests with correction for multiple testing using the Benjamini, Krieger, and Yekutieli procedure (FDR < 0.01); *p*-values in Supplementary Data 4. Bar plots show mean, error bars = +1 S.D. Source data are provided as a Source Data file.

U6 C60 2′-*O*-methylation, whereas the mutant construct (mutOE) did not rescue U6 methylation (Fig. 2C). Next, we assessed the in vitro phenotypes of these stably transfected cell lines. Compared to Snord67KO-2 cells transfected with empty vector, cells transfected with the wild-type Snord67 expression construct exhibited increased colony formation, while cells transfected with the mutant Snord67 construct showed partial rescue of colony formation (Fig. 2D). In addition to showing decreased colony formation, the Snord67KO-2 cell line also formed smaller spheroids in a 3-dimensional growth assay for assessing tumorigenesis (Fig. 2E). Stable rescue of wild-type Snord67 expression led to increased spheroid area, similar to the 4T1 WT cells, whereas mutant Snord67 did not have as much of an effect on spheroid area (Fig. 2E). These results demonstrate that expression of wild-type Snord67 is sufficient to rescue the colony and spheroid formation phenotypes of Snord67 knockout 4T1 cells, while expression of mutant Snord67 partially rescues these phenotypes.

To replicate these findings in another model, we generated a Snord67 single-cell knockout clone in EO771.LMB, another murine triple-negative breast cancer model that spontaneously metastasizes to lymph nodes in vivo[32]. Loss of Snord67 expression was confirmed by RT–qPCR (Fig. 3A). CKAP5 mRNA expression and splicing in the region flanking Snord67 were not affected by Snord67 knockout (Supplemental Fig. 5A, B), while CKAP5 protein expression was only mildly decreased upon Snord67 knockout (Supplemental Fig. 5C). Snord67 knockout cells had markedly decreased U6 C60 2′-*O*-methylation levels compared to wildtype, as measured by Nm-VAQ (Supplemental Fig. 5D), but Snord67 knockout had no effect on U6 expression (Supplemental Fig. 5E). Similar to our findings in 4T1 cells (Fig. 1F–H), loss of Snord67 in EO771.LMB cells resulted in significantly decreased proliferation and colony formation (Fig. 3B, C). To test whether restoring Snord67 expression is sufficient to rescue these phenotypes, EO771.LMB WT and Snord67 knockout cells were transfected with in vitro transcribed murine Snord67 or with a synthetic control snoRNA (human Snord75), or were treated with transfection reagent only. Whereas transfection of 4T1 cells with in vitro transcribed Snord67 resulted in only transient rescue of Snord67 levels, transfection of EO771.LMB cells with in vitro transcribed Snord67, but not with a control snoRNA, resulted in sustained rescue of Snord67 expression for up to 5 days (Fig. 3D) and was sufficient to significantly rescue U6 C60 2′-*O*-methylation in Snord67 knockout cells (Fig. 3E). Compared to treatment with transfection reagent only or transfection with control snoRNA, transfection with in vitro transcribed murine Snord67 resulted in phenotypic rescue of colony formation in EO771.LMB Snord67 knockout cells (Fig. 3F). Thus, the EO771.LMB murine breast cancer cell line exhibits in vitro phenotypes similar to the 4T1 cell line upon Snord67 knockout and rescue.

To replicate these findings in a human breast cancer model, we generated a Snord67 knockout clone in MDA-MB-231-D3H2LN-luc (D3H2), a cell line that spontaneously metastasizes to loco-regional

lymph nodes and was derived from the human triple-negative breast adenocarcinoma model MDA-MB-231[33]. Loss of Snord67 expression was verified by RT–qPCR (Fig. 3G). CKAP5 mRNA expression and splicing in the region flanking Snord67, as well as CKAP5 protein expression, were not affected by Snord67 knockout (Supplemental Fig. 6A–C). Consistent with our findings in murine cell lines, loss of Snord67 in D3H2 cells resulted in markedly decreased 2′-*O*-methylation at the C60 position in U6 snRNA as measured by Nm-VAQ (Fig. 3H). The D3H2 cell line did not readily form colonies on plastic, so we used a spheroid assay to measure tumorigenesis in vitro. Similar to our findings in 4T1 cells (Fig. 2E), the D3H2 Snord67 knockout cells formed smaller spheroids than the D3H2 WT cells (Fig. 3I). As a complementary approach to knocking down Snord67 expression, we designed an antisense oligonucleotide (ASO) targeting the antisense element of Snord67. This second-generation ASO was chemically-modified with flanking 2′-methoxyethyl (MOE) bases for enhanced stability, resistance to nuclease degradation, and avoidance of immunogenicity (Supplemental Fig. 6D). Transfection of D3H2 WT cells with Snord67 ASO led to sustained, dose-dependent knockdown of Snord67 (Supplemental Fig. 6E) and decreased 2′-*O*-methylation at the C60 position in U6 snRNA (Supplemental Fig. 6F) without affecting host gene expression (Supplemental Fig. 6G, H). Similar to our findings in Snord67 knockout cells compared to D3H2 WT cells (Fig. 3I), knockdown of Snord67 with an ASO led to the formation of smaller spheroids compared to transfection with a control ASO (Supplemental Fig. 6I).

Taken together, our findings demonstrate that Snord67 is required for 2′-*O*-methylation at C60 in U6 snRNA and that Snord67 promotes colony and spheroid formation in murine and human triple-negative breast cancer cell lines. We also showed in murine models that expression of wild-type Snord67 rescued both 2′-*O*-methylation of U6 and colony and spheroid formation, whereas expression of mutant Snord67 did not rescue 2′-*O*-methylation of U6 and only partially rescued colony and spheroid formation.

## Snord67 is necessary for LN tumor growth and distant metastasis

To evaluate the effect of Snord67 on LN tumor growth and distant metastasis in vivo, we micro-injected 4T1 WT or Snord67 knockout cells expressing mCherry and Renilla luciferase into the AxLN of immune-competent mice (Fig. 4A). Mice that were injected with Snord67 knockout cells exhibited significantly decreased growth of AxLN tumors (Fig. 4B). Using an assay to detect luciferase activity in cancer cells that had metastasized to the lungs, we found that mice injected with Snord67 knockout cells also developed significantly fewer distant lung metastases four weeks after AxLN injection (Fig. 4C). Since the observed tumor growth phenotypes could be due to the isolation of clonal populations of Snord67 knockout cells, we also used an alternative approach by abruptly inhibiting Snord67 expression

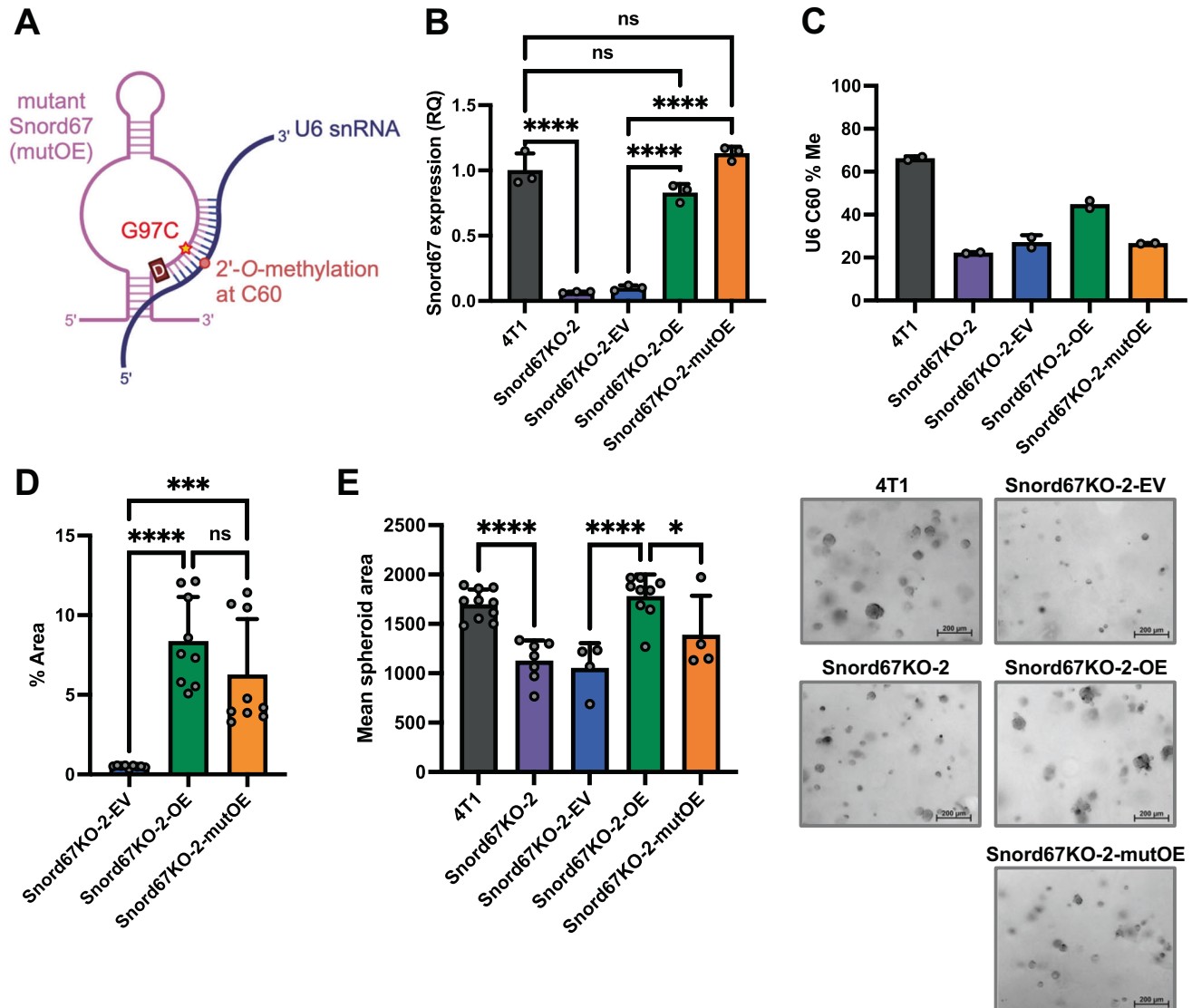

**Fig. 2 | Colony and spheroid formation upon stable rescue of Snord67 in 4T1 cells. A** Schematic of the Snord67 mutant construct mutOE, which contains the mutation G97C at the nucleotide complementary to the target site C60 in U6 snRNA. Created in BioRender. Zhou, K. (2025) https://BioRender.com/v01z684. **B** Expression of Snord67 in 4T1 WT cells (4T1), Snord67 knockout cells (Snord67KO-2), and Snord67KO-2 cells stably transfected with empty vector (EV), Snord67 expression construct (OE), or mutant Snord67 expression construct (mutOE), quantified by RT–qPCR and presented as relative quantification (RQ) normalized to GAPDH expression and to 4T1 WT cells. $n = 3$ biological replicates per cell line. Statistical significance was calculated by one-way ANOVA using the Šidák method for multiple comparisons; ns = not significant, ****adjusted $p < 0.0001$; adjusted $p < 10^{-6}$ for 4T1 vs. Snord67KO-2, Snord67KO-2-EV vs. Snord67KO-2-OE, and Snord67KO-2-EV vs. Snord67KO-2-mutOE; adjusted $p = 0.065$ for 4T1 vs. Snord67KO-2-OE; adjusted $p = 0.22$ for 4T1 vs. Snord67KO-2-mutOE. **C** Percent methylation of U6 snRNA at the C60 position (U6 C60 % Me) in 4T1 WT cells, Snord67KO-2 cells, and stably transfected Snord67KO-2 cells, quantified by Nm-VAQ. n = 2 biological replicates per cell line. **D** Colony formation assay at day 5 in stably transfected Snord67KO-2 cells, quantified as percent area. $n = 9$ replicates per cell line. Statistical significance was calculated by one-way ANOVA using the Šidák method for multiple comparisons; ns = not significant, ***$p < 0.001$, ****$p < 0.0001$. Adjusted $p < 0.0001$ for Snord67KO-2-EV vs. Snord67KO-2-OE; adjusted $p = 0.0002$ for Snord67KO-2-EV vs. Snord67KO-2-mutOE; adjusted $p = 0.27$ for Snord67KO-2-OE vs. Snord67KO-2-mutOE. **E** Spheroid assay at day 3 in 4T1 WT, Snord67KO-2, and stably transfected Snord67KO-2 cells. Mean spheroid area was quantified using 4–10 images per cell line: 4T1 (10), Snord67KO-2 (7), Snord67KO-2-EV (4), Snord67KO-2-OE (9), Snord67KO-2-mutOE (4). Scale bar = 200 μm. Statistical significance was calculated by one-way ANOVA using the Šidák method for multiple comparisons; ns = not significant, *adjusted $p < 0.05$, ****adjusted $p < 0.0001$. Adjusted $p < 0.0001$ for 4T1 vs. Snord67KO-2 and Snord67KO-2-EV vs. Snord67KO-2-OE; adjusted $p = 0.032$ for Snord67KO-2-OE vs. Snord67KO-2-mutOE. All bar plots show mean values, with error bars representing +1 S.D. Source data are provided as a Source Data file.

with an ASO. In vitro treatment of 4T1 cells with Snord67 ASO or Snord111 ASO resulted in significantly decreased Snord67 or Snord111 expression, respectively (Supplemental Fig. 7A, B), but only treatment with Snord67 ASO led to significantly decreased colony formation (Supplemental Fig. 7C). To test the effects of Snord67 suppression with ASO in vivo, 4T1 WT cells were micro-injected into AxLNs to generate tumors. Two weeks after injection, mice were randomized to the following treatment groups: 1) phosphate-buffered saline (PBS); 2)

negative control ASO at 48 mg/kg (mpk); and 3) Snord67 ASO at 12 mpk, 4) 24 mpk, or 5) 48 mpk. Baseline AxLN tumors were measured by caliper, and ASO was administered daily subcutaneously for six consecutive days (Fig. 4D). AxLN tumors were measured and harvested 24 hours after the last ASO treatment, and Snord67 knockdown was confirmed by RT–qPCR (Fig. 4E). Mice treated with 24 and 48 mpk of Snord67 ASO demonstrated significantly reduced tumor growth at 20 days post-LN injection, compared to vehicle-treated mice (Fig. 4F).

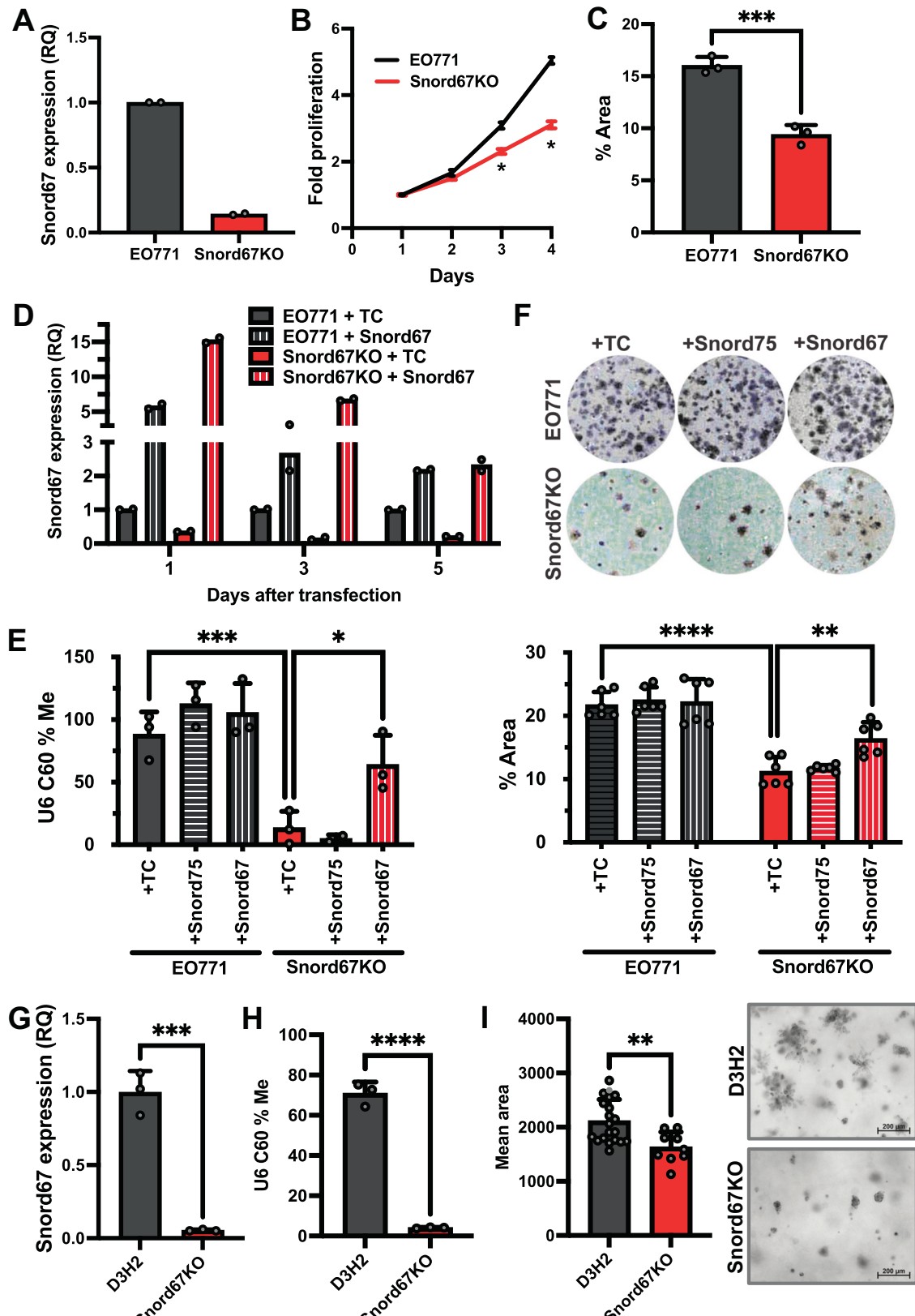

Together, these results demonstrate that loss of Snord67 expression, whether by CRISPR knockout or by ASO treatment, leads to decreased AxLN tumor growth in a microsurgical, immune-competent murine model of LN metastasis. Although the effect of Snord67 ASO on distant metastases was not assessed due to the short duration of follow-up, our findings in two Snord67 knockout clones further suggest that loss

of Snord67 leads to decreased distant metastases arising from micro-injected AxLN tumors.

Next, we interrogated the role of Snord67 along different steps of the metastatic cascade. To evaluate whether Snord67 may play a role in distant metastasis colonization after the cancer cells leave the LN and enter the circulation, we tested the effect of Snord67 loss on the

**Fig. 3 | Knockout and rescue of Snord67 expression in EO771.LMB and D3H2 cells. A** Relative quantification (RQ) of Snord67 expression in EO771.LMB WT cells (EO771) and Snord67 knockout cells (Snord67KO) by RT−qPCR, $n = 2$ technical replicates, repeated independently at least 3 times with similar results. **B** Cell proliferation assay of EO771 and Snord67KO cells, measured by absorbance following addition of alamarBlue cell viability reagent (line=mean, n = 3 biological replicates). Statistical significance (*) was calculated by multiple unpaired t-tests with correction for multiple testing using the Benjamini, Krieger, and Yekutieli method (FDR < 0.05): $p = 0.024$ (3 days), $p = 0.0057$ (4 days). **C** Colony formation assay of EO771 and Snord67KO cells quantified by area of well coverage, plated in 3 replicates. ***$p < 0.001$; $p = 0.0007$ by two-tailed t-test. **D** Snord67 expression by RT−qPCR at 1, 3, and 5 days after transfection of EO771 and Snord67KO cells with Snord75 or Snord67. $n = 2$ technical replicates, repeated independently 4 times with similar results. **E** Measurement of U6 C60 2'-O-methylation (U6 C60 % Me) in cells transfected with transfection reagent only (TC), Snord75, or Snord67 by Nm-

VAQ. n = 3 biological replicates (except $n = 2$ for Snord67KO + Snord75). $p = 0.0009$ (EO771 + TC vs. Snord67KO+TC), $p = 0.0136$ (Snord67KO+Snord67 vs. Snord67KO +TC) by one-way ANOVA; *$p < 0.05$, ***$p < 0.001$. **F** Colony formation assay after transfection with synthetic snoRNAs. $n = 6$ biological replicates. $p < 0.0001$ (EO771 + TC vs. Snord67KO+TC), $p = 0.0012$ (Snord67KO+Snord67 vs. Snord67KO +TC) by one-way ANOVA; **$p < 0.01$, ****$p < 0.0001$. **G** RQ of Snord67 in MDA-MB-231-D3H2LN-luc WT cells (D3H2) and Snord67 knockout cells (Snord67KO) by RT−qPCR, normalized to GAPDH. $n = 3$ biological replicates. $p = 0.0004$ by two-tailed t-test; ***$p < 0.001$. **H** U6 C60 % Me in D3H2 WT and Snord67 knockout cells. n = 3 biological replicates. $p < 0.0001$ by two-tailed t-test; ****$p < 0.0001$. **I** Mean spheroid area at day 7 in D3H2 WT cells and Snord67KO cells, quantified using 19 (D3H2) and 9 (Snord67KO) images. Scale bar = 200 μm. $p = 0.0029$ by two-tailed t-test; **$p < 0.01$. Bar plots show mean, error bars = +1 S.D. Source data are provided as a Source Data file.

---

establishment of metastases after tail vein injection (Fig. 4G). Consistent with reduced spontaneous lung metastases from LNs (Fig. 4C), we observed a dramatic reduction in lung metastases in mice that underwent tail vein injection with either of the Snord67 knockout clones compared to tail vein injection with 4T1 WT cells (Fig. 4H). We then evaluated the effect of Snord67 expression on growth and metastasis from the primary tumor. First, 4T1 and EO771.LMB WT and Snord67 knockout cells were injected into the mammary fat pads of mice (Fig. 5A). LN and lung tissues were mixed with known quantities of 4T1 and EO771.LMB cells to generate a standard curve for quantifying cell number using qPCR detection of the mCherry reporter gene (Supplemental Fig. 8). Then, four weeks after cancer cells were injected in the MFP to form tumors, axillary LNs and lungs were harvested, and qPCR of mCherry expression was performed to quantify cancer cells that had metastasized to LN and lungs. Both 4T1 Snord67 knockout clones and the EO771.LMB Snord67 knockout clone demonstrated significantly decreased LN metastases, suggesting that Snord67 plays an important role in promoting metastasis to lymph nodes (Fig. 5B, C). However, in contrast to LN tumor growth after AxLN micro-injection (Fig. 4B), MFP tumor growth was not clearly affected by loss of Snord67. Although the EO771.LMB Snord67 knockout clone and one 4T1 Snord67 knockout clone demonstrated significantly decreased MFP tumor growth after mammary fat pad injection, MFP tumor growth was unaffected in the second 4T1 Snord67 knockout clone (Fig. 5D, E). Additionally, in contrast to the development of lung metastases from AxLN tumors (Fig. 4C), the development of lung metastases arising from MFP tumors was not consistently affected by loss of Snord67, since decreased lung metastases were only observed with one 4T1 Snord67 knockout clone (Fig. 5F, G). Of note, the routes of metastasis differ between lung metastases that arise from MFP tumors (Fig. 5F, G) and lung metastases that arise from AxLN tumors (Fig. 4C). For the former, the predominant route is hematogenous dissemination, while for the latter, the predominant route is lymphatic dissemination. Since Snord67 knockout had a more consistent impact on lung metastases arising from AxLN tumors than on lung metastases arising from MFP tumors, Snord67 may play a more important role in promoting metastasis by the lymphatic route and a lesser role in promoting metastasis by the hematogenous route. Together, these results demonstrate that Snord67 decreases de novo LN metastases but does not clearly affect MFP tumor growth or distant metastases arising from the MFP, suggesting that the role of Snord67 on tumor growth and metastasis may be specific to the LN environment.

## Snord67 knockout leads to changes in the splicing landscape

To explore mechanisms by which Snord67 could contribute to metastatic progression, we used RNA sequencing (RNA-seq) to examine differential gene expression in 4T1 WT cells compared to two Snord67 knockout clones. We identified 3,822 genes that were significantly differentially expressed in Snord67KO-1 cells compared to 4T1

WT cells and 2,131 genes that were significantly differentially expressed in Snord67KO-2 cells compared to 4T1 WT cells (Supplemental Fig. 9A, B). There was a significant overlap of 1,301 genes that were differentially expressed in both Snord67 knockout clones relative to 4T1 WT cells ($p < 10^{-12}$ by the hypergeometric test; Supplemental Fig. 9C). Moreover, the two Snord67 knockout clones exhibited a significant correlation in the magnitude and direction of change in gene expression levels relative to 4T1 WT cells (Supplemental Fig. 9D). Biological processes that were enriched among the 1,301 overlapping differentially expressed genes included extracellular matrix organization and cytokine signaling (Supplemental Fig. 9E). Since both remodeling of the extracellular matrix and dysregulation of cytokine signaling have been implicated in tumor progression and metastasis[34,35], Snord67 may promote metastasis in part by regulating the expression of genes involved in these pathways.

Because Snord67 guides 2'-O-methylation of the core spliceosome component U6 snRNA, we hypothesized that loss of Snord67 could lead to fundamental changes in splicing, which may contribute to the Snord67 knockout phenotypes. We therefore evaluated the RNA-seq data for changes in splicing patterns between 4T1 WT and Snord67 knockout cells. Alternative splicing events were considered differentially spliced if their percent spliced in (PSI) value changed by more than 10% ($|\Delta PSI| \geq 0.1$) between 4T1 WT and Snord67 knockout cells, and were considered statistically significant at a threshold false discovery rate (FDR) of 5% (FDR < 0.05). We identified 1,624 significant differential alternative splicing events in 1158 genes in Snord67KO-1 cells compared to 4T1 WT cells and 1,870 significant differential alternative splicing events in 1,380 genes in Snord67KO-2 cells compared to 4T1 WT cells. Most differential alternative splicing events between 4T1 WT and Snord67 knockout cells were categorized as cassette exons (CE) or mutually exclusive exons (MXE) (Fig. 6A–C). Of 787 genes with differential CE events in Snord67KO-1 cells compared to 4T1 WT cells and 504 genes with differential CE events in Snord67KO-2 cells compared to 4T1 WT cells, there was a significant overlap of 251 genes with differential CE events in both Snord67 knockout clones relative to 4T1 WT cells ($p < 10^{-12}$ by the hypergeometric test; Fig. 6D). Reverse transcription and PCR (RT−PCR) followed by polyacrylamide gel electrophoresis was used to validate several of these alternative splicing events (Supplemental Fig. 10A–C). Gene ontology analysis showed that genes with differential CE events encoded proteins involved in RNA splicing and regulation of mRNA processing (Fig. 6E–F). Intron retention was the next most common category of differential splicing event observed in Snord67 knockout cells compared to 4T1 WT cells (Fig. 6A). Intron retention may serve as a proxy for splicing efficiency, in which a higher intron retention ratio corresponds to a decrease in splicing efficiency. Therefore, we assessed the effect of loss of Snord67 on global splicing efficiency by measuring the intron retention ratio (reads mapping to introns vs. exons, Methods) for each intron in 4T1 WT cells, Snord67KO-1 cells,

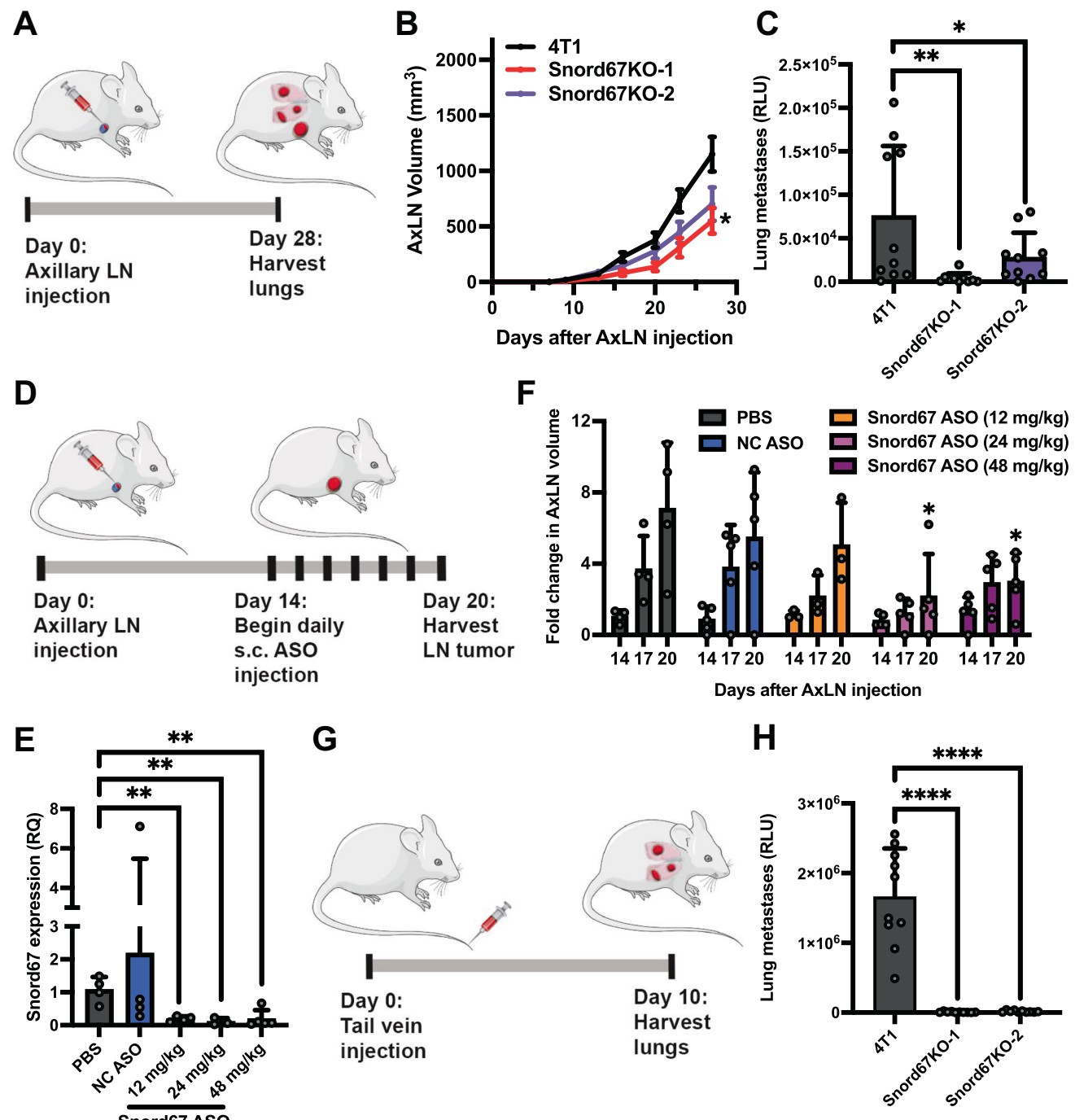

**Fig. 4 | Impact of Snord67 loss on AxLN tumor growth and metastasis.**
**A** Schematic of microsurgical injection of axillary lymph node (AxLN) leading to subsequent lung metastases. **B** AxLN tumor volumes by calipers (line = mean, error bars = SEM, n = 10 mice/group). Statistical significance was calculated by multiple unpaired t-tests with correction for multiple testing using the Benjamini, Krieger, and Yekutieli method (FDR < 0.05); *adjusted $p < 0.01$ for Snord67KO-1 vs. 4T1 from 9–27 days ($p$-values in Supplementary Data 4). **C** Quantification of lung metastases at day 28 after AxLN injection. Harvested lungs were digested into a single cell suspension and Relative Luciferase Units (RLU) measured to quantify lung metastases. $n = 10$ mice per group, 3 technical replicates per mouse. $p = 0.0031$ (Snord67KO-1 vs. 4T1), $p = 0.0394$ (Snord67KO-2 vs. 4T1) by one-way ANOVA; *$p < 0.05$, **$p < 0.01$. **D** Schematic of AxLN injection to form tumors, followed by daily subcutaneous (s.c.) treatment with antisense oligonucleotide (ASO). AxLN tumors were harvested at day 20 following injection of tumor cells. **E** Relative quantification (RQ) of Snord67 by RT–qPCR in AxLN tumors following treatment with ASO, normalized to GAPDH and RPLP0 expression. Dose of negative control

(NC) ASO was 48 mg/kg. n = 3 (24 mg/kg), 4 (PBS, NC ASO, 12 mg/kg), or 5 (48 mg/kg) biological replicates. $p = 0.0044$, 0.0094, 0.0047 (PBS vs. 12, 24, 48 mg/kg) by two-tailed t-tests without adjustment. **F** AxLN tumor volumes by calipers prior to ASO treatment, during ASO treatment, and on day of AxLN tumor harvest. n = 5 mice per treatment. $p = 0.0225$ and 0.0295 (PBS vs. 24 and 48 mg/kg) by unpaired one-tailed t-tests without adjustment. **G** Schematic of experimental metastasis: injection of cancer cells into the tail veins of mice, with subsequent metastasis to the lungs. **H** Quantification of lung metastases 10 days after tail vein injection. Harvested lungs were mechanically and enzymatically digested, cells were lysed, and RLU were measured. n = 10 mice/group, 3 technical replicates per mouse. $p < 0.0001$ for both comparisons by one-way ANOVA (***$p < 0.0001$). Bar plots show means, error bars = +1 S.D. The diagrams in (**A**, **D**, **G**) are modified from Servier Medical Art (https://smart.servier.com/smart_image/mouse; https://smart.servier.com/smart_image/lungs-11) and Pngimg.com (https://pngimg.com/image/12387). Source data are provided as a Source Data file.

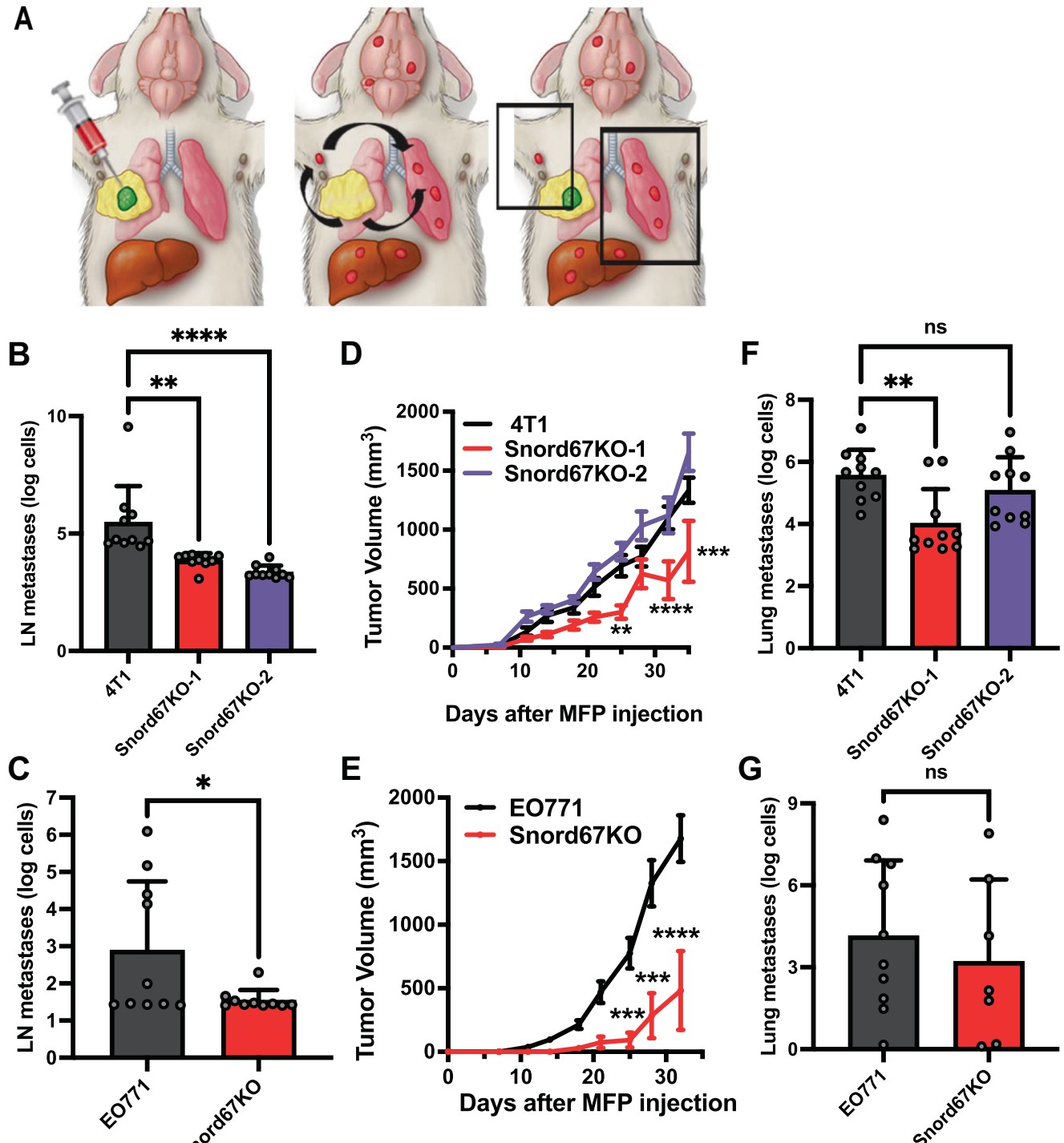

**Fig. 5 | Impact of Snord67 knockout on LN and lung metastases after MFP injection. A** Schematic of MFP injection and detection of subsequent LN and lung metastases. © 2020 The University of Texas M.D. Anderson Cancer Center. Quantification of 4T1 (**B**) and EO771.LMB (**C**) LN metastases after MFP injection by qPCR detection of mCherry reporter gene expression in tumor-draining AxLNs harvested 28 days after MFP injection. n = 10 mice per group. Statistical significance was calculated by one-way ANOVA; ns, not significant; **$p < 0.01$, ****$p < 0.0001$. For (**B**), $p = 0.0015$ for Snord67KO-1 vs. 4T1, and $p < 0.0001$ for Snord67KO-2 vs. 4T1. For (**C**), $p = 0.0468$ for Snord67KO vs. EO771. Measurement of 4T1 (**D**) and EO771.LMB (**E**) MFP tumor volumes measured by calipers twice weekly. n = 10 mice. Mean ± 1 S.E.M. is shown. Statistical significance was calculated by two-way ANOVA with Dunnett's method (**D**) or the Šidák method (**E**) for multiple comparisons;

**$p < 0.01$, ***$p < 0.001$, ****$p < 0.0001$. For (**D**), $p$ values for Snord67KO-1 vs. 4T1 were 0.0040, < 0.0001, and 0.0001 at 25, 32, and 35 days, respectively. For (**E**), $p$-values for Snord67KO vs, EO771 were 0.0002, < 0.0001, and < 0.0001 at 25, 28, and 32 days, respectively. Quantification of 4T1 (**F**) and EO771.LMB (**G**) lung metastases after MFP injection, expressed as the logarithm of the number of tumor cells as determined based on qPCR detection of mCherry. n = 10 mice per group (except n = 7 mice for EO771 Snord67KO group). Statistical significance was calculated by two-tailed t-test; ns, not significant; *$p < 0.05$. For (**F**), $p = 0.0050$ for Snord67KO-1 vs. 4T1, and $p = 0.5285$ for Snord67KO-2 vs. 4T1. For (**G**), $p = 0.5240$ for Snord67KO vs. EO771. All bar plots show mean values, with error bars representing +1 S.D. Source data are provided as a Source Data file.

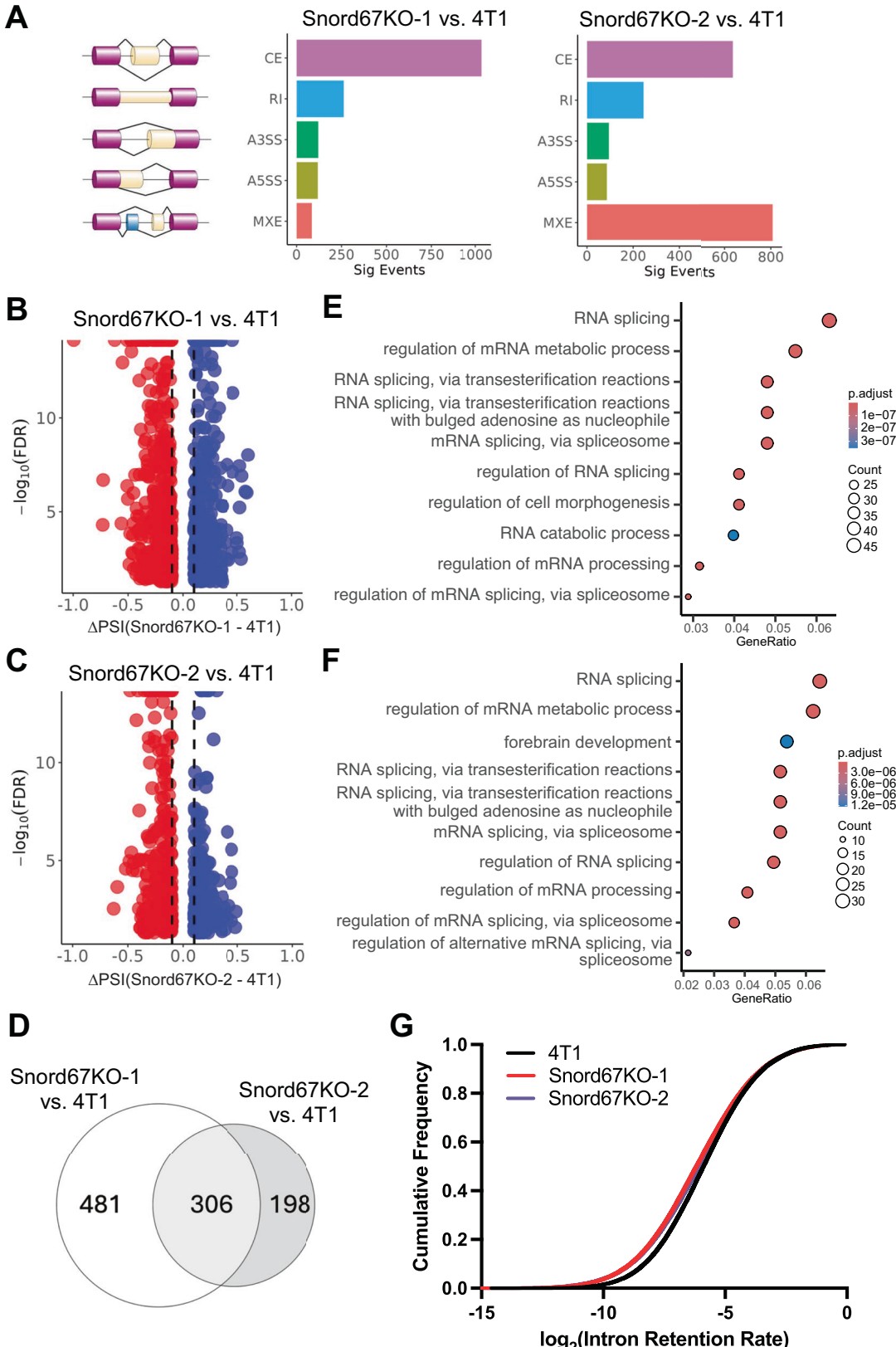

and Snord67KO-2 cells. Snord67KO-1 and Snord67KO-2 cells exhibited lower intron retention ratios and thus higher splicing efficiency compared to 4T1 WT cells (Fig. 6G and Supplemental Fig. 10D), and these differences in intron retention ratios were significant by the Kolmogorov-Smirnov test (Snord67KO-1 vs. 4T1 WT, $p < 0.0001$; Snord67KO-2 vs. 4T1 WT, $p < 0.0001$). Together, these results show

that loss of Snord67 results in widespread changes to splicing, marked by a reduction in intron retention that implies increased splicing efficiency.

We further examined the impact of Snord67 knockout on $2'$-$O$-methylation in 4T1 cells using ribose oxidation sequencing (RibOxi-seq). In RibOxi-seq, RNA is randomly fragmented and then oxidized

**Fig. 6 | Changes in the splicing landscape upon Snord67 knockout in 4T1 cells.**
**A** Distribution of types of alternative splicing events (Sig Events) in Snord67KO-1 and Snord67KO-2 cells compared to 4T1 WT cells. CE cassette exon, RI retained intron, A3SS alternative 3′-splice site, A5SS alternative 5′-splice site, MXE mutually exclusive exons. Splicing diagrams created in Adobe Illustrator. **B, C** Volcano plots of differential CE events with |ΔPSI| ≥ 0.1 and FDR < 0.05 in Snord67KO-1 and Snord67KO-2 cells compared to 4T1 WT cells. Exons with significantly increased inclusion in Snord67 knockout clones relative to 4T1 WT cells are shown in blue, while exons with significantly decreased inclusion in Snord67 knockout clones are shown in red. PSI percent spliced in, FDR false discovery rate. **D** Venn diagram showing the overlap between genes with differential CE events in Snord67KO-1 cells and Snord67KO-2 cells compared to 4T1 WT cells. **E, F** Gene ontology analysis of genes with differential CE events in 4T1 WT cells versus Snord67 knockout cells. Enrichment analysis was performed in R using the enrichGO function with a $p$-value cut-off of < 0.05, and the Benjamini-Hochberg procedure was applied to adjust for multiple testing. Individual $p$-values are provided in Supplementary Data 4. **G** Cumulative distribution of intron retention ratio in 4T1 WT cells, Snord67KO-1 cells, and Snord67KO-2 cells. Statistical significance was determined by Kolmogorov-Smirnov test (Snord67KO-1 vs. 4T1 WT, $p < 0.0001$; Snord67KO-2 vs. 4T1 WT, $p < 0.0001$).

such that fragments with 2′-*O*-methylated nucleotides at the 3′ end are protected, while unmethylated nucleotides are degraded[36,37]. Fragments ending in methylated nucleotides are positively selected and enriched during library construction, and sequencing of the resulting libraries allows for the mapping of 2′-*O*-methylation sites. Since Snord67 guides 2′-*O*-methylation of C60 in U6, we first examined 2′-*O*-methylation patterns on U6 snRNA in our RibOxi-seq data. In 4T1 WT cells, all eight known 2′-*O*-methylation sites on U6 snRNA were detected. In Snord67 knockout cells, there was a loss of methylated reads specifically at site C60, consistent with our Nm-VAQ experiments showing that Snord67 is essential for 2′-*O*-methylation of C60 in U6 snRNA (Supplemental Fig. 11A). Next, we considered whether RibOxi-seq might reveal non-canonical targets of Snord67-guided 2′-*O*-methylation. Specifically, since we observed many differential alternative splicing events in Snord67 knockout cells, we wondered whether the differentially spliced pre-mRNAs contained Snord67-guided 2′-*O*-methylation sites that modulate alternative splicing independent of U6 modification. However, other than C60 in U6 snRNA, we did not identify any sites that were differentially 2′-*O*-methylated in Snord67 knockout cells compared to 4T1 WT cells by RibOxi-seq. In particular, no Snord67-dependent 2′-*O*-methylation sites were identified in genes that were differentially spliced in Snord67 knockout cells compared to 4T1 WT cells (Supplemental Fig. 11B–G). Therefore, we were not able to identify any additional 2′-*O*-methylation targets of Snord67 besides C60 in U6 snRNA, although we cannot exclude the existence of other Snord67-dependent 2′-*O*-methylation sites that were not detected at the sensitivity level of RibOxi-seq.

To examine the impact of Snord67 knockout on gene expression and splicing patterns in a human breast cancer cell line, we performed RNA-seq in D3H2 WT cells and Snord67 knockout cells. We identified 422 upregulated and 232 downregulated genes in Snord67 knockout cells compared to D3H2 WT cells (Fig. 7A). Gene ontology analysis of differentially regulated genes showed enrichment for genes involved in cytokine production and cell adhesion (Fig. 7B). Notably, dysregulation of cytokines can alter the tumor microenvironment and contribute to metastasis, while disruption of cell adhesion is critical for both malignant transformation and metastasis[34,38]. We also identified 966 differential alternative splicing events in 745 genes in D3H2 WT cells compared to Snord67 knockout cells, of which the majority were cassette exon events (Fig. 7C, D). Gene ontology analysis showed that genes with differential CE events encoded proteins that function in cell division and DNA repair (Fig. 7E). Using RT–PCR, we validated that exon 40 of the *MYO18A* gene, exon 3 of the *NFYA* gene, and exon 25 of the *FN1* gene are differentially spliced in D3H2 WT cells compared to Snord67 knockout cells (Fig. 8A, B and Supplemental Fig. 12A). Interestingly, alternatively spliced isoforms of *MYO18A* and *NFYA* have been linked to prognosis in breast cancer patients[39,40], while *FN1* is a critical component of the extracellular matrix that has been implicated in tumor progression and metastasis[41].

Taken together, these results demonstrate that loss of Snord67 leads to changes in gene expression and, more notably, widespread changes in the splicing landscape, including effects on alternative splicing of cassette exons as well as effects on overall splicing

efficiency. Although we did not identify any additional 2′-*O*-methylation targets of Snord67, our RibOxi-seq results provide further evidence that Snord67 is required for the 2′-*O*-methylation of C60 in U6 snRNA.

## Snord67 is associated with differential alternative splicing in breast cancer patients

Since Snord67 knockout was associated with differential alternative splicing in both murine and human breast cancer cell lines (Figs. 6–7), we considered whether these same alternative splicing events correlated with Snord67 expression in humans with breast cancer. We performed RNA-seq of 24 matched pairs of primary breast tumors and LN metastases from patients with breast cancer. Since snoRNAs are largely undetected in poly(A)-selected RNA-seq libraries, we measured Snord67 expression in the same specimens by RT–qPCR. Of the three alternative splicing events validated by RT–PCR in D3H2 cells (Fig. 8A, B and Supplemental Fig. 12), two events, involving exon 40 of the *MYO18A* gene and exon 3 of the *NFYA* gene, significantly correlated with Snord67 expression in the primary breast and LN tumor specimens from breast cancer patients (*MYO18A* exon 40: Pearson $r = 0.46$, $p = 0.0009$; *NFYA* exon 3: Pearson $r = 0.49$, $p = 0.0006$; Fig. 8C, D). These two exons also exhibited significantly different exon inclusion rates in tumors with low Snord67 expression compared to those with high Snord67 expression by t-test (*MYO18A* exon 40: threshold −ΔCt(Snord67/Gapdh) = −1.83, $p = 0.0005$; *NFYA* exon 3: threshold −ΔCt(Snord67/Gapdh) = −1.465, $p = 0.0015$; Fig. 8C–D). Moreover, the change in PSI for these two alternative exons significantly correlated with the change in Snord67 expression in matched pairs of primary breast tumors and LN metastases (*MYO18A* exon 40: Pearson $r = 0.54$, $p = 0.0066$; *NFYA* exon 3: Pearson $r = 0.51$, $p = 0.011$; Fig. 8C, D). That is, in patients whose LN metastasis displayed increased Snord67 expression relative to the primary breast tumor, the inclusion of *MYO18A* exon 40 and *NFYA* exon 3 was also increased in the LN tumor relative to the primary breast tumor. Taken together, these results support a model in which differential expression of Snord67 leads to differential alternative splicing not only in murine and human breast cancer cell lines, but also in patients with breast cancer.

## Snord67 expression is associated with subtype and prognosis in breast cancer patients

To more directly evaluate the relevance of Snord67 expression to patients with breast cancer, we examined data from breast cancer patients in The Cancer Genome Atlas (TCGA). Although mRNA sequencing does not accurately measure snoRNA expression, a previous study developed the SNORic dataset, which measured snoRNA expression in human cancers by mapping reads from TCGA microRNA sequencing (miRNA-seq) data to snoRNAs[11]. By integrating the SNORic database with breast cancer subtypes and updated clinical outcomes, we found that Snord67 expression was significantly increased in the basal and luminal B subtypes, which are among the most clinically aggressive subtypes of breast cancer, compared to the luminal A subtype, which is associated with better prognosis (Supplemental Fig. 12B). Moreover, high Snord67 expression was associated with

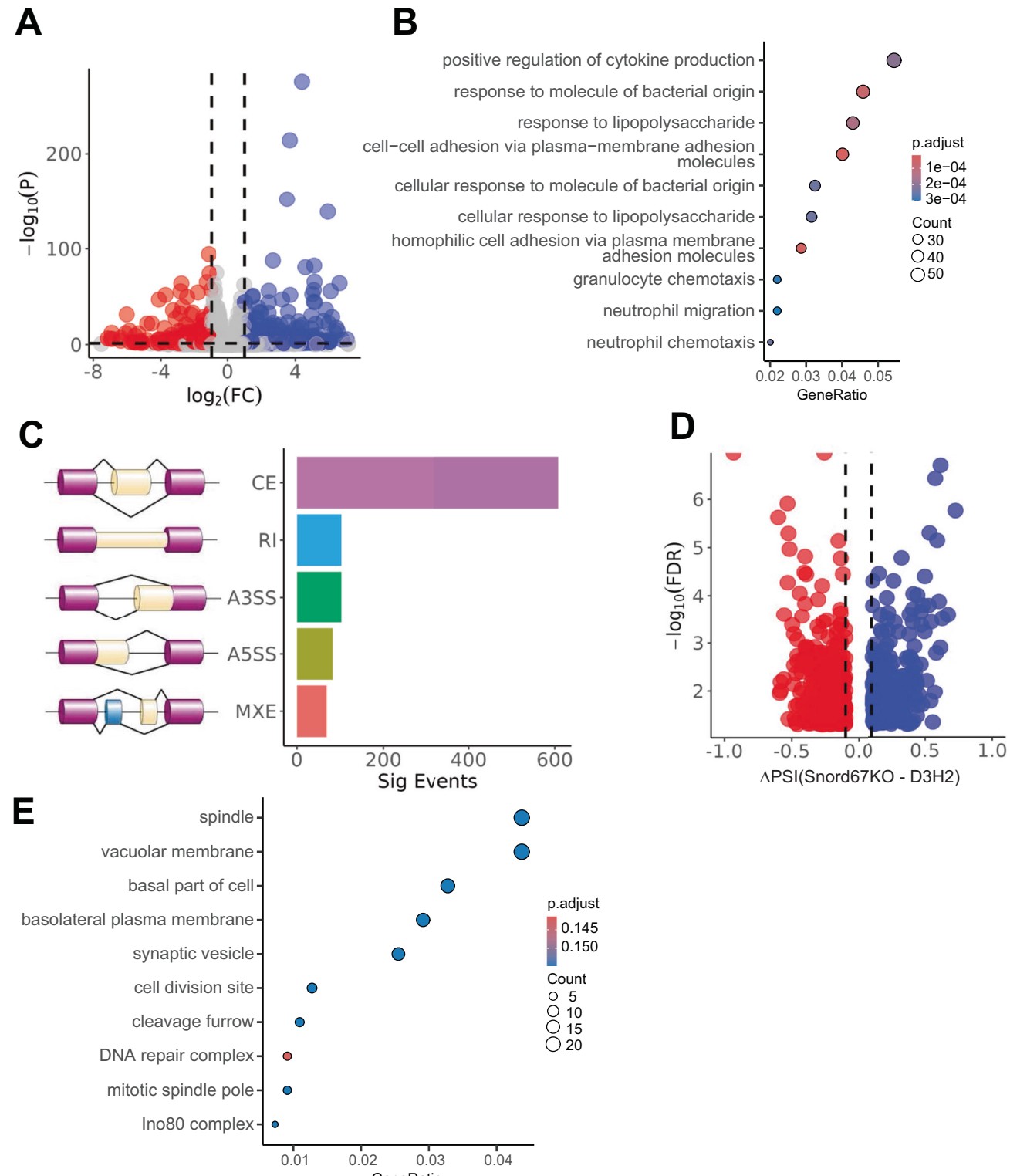

significantly decreased overall survival in the luminal B subtype (Supplemental Fig. 12C). In the HER2-enriched subtype of breast cancer, tumors with high Snord67 expression trended toward decreased overall survival, but this difference was not statistically significant (Supplemental Fig. 12D). We observed no significant associations of Snord67 expression with survival in Luminal A or basal subtypes. Interestingly, high expression of the Snord67 host gene *CKAP5* was associated with significantly worse overall survival in HER2-enriched breast cancer (Supplemental Fig. 12E), consistent with previous results

showing that *CKAP5* expression is associated with poor prognosis in non-small cell lung cancer[42]. Notably, Snord67 and *CKAP5* expression were highly correlated in breast cancer patients (Supplemental Fig. 12F). Although we have shown in several models that loss of Snord67 impacted phenotypes independent of *CKAP5* host gene expression, the correlation of Snord67 and *CKAP5* expression in breast cancer patients raises the possibility that Snord67 expression may be co-regulated with expression of its host gene. Considering this relationship, we evaluated whether high expression levels of both Snord67

**Fig. 7 | Differential gene expression and alternative splicing upon Snord67 knockout in D3H2 cells. A** Volcano plot of differentially expressed genes with | log₂(foldchange)| > 1 and an adjusted *p*-value < 0.05 in Snord67KO cells compared to MB-231-D3H2LN-luc WT cells (D3H2). Significantly upregulated genes are in blue, and significantly downregulated genes are in red. **B** Gene ontology analysis of genes that were differentially expressed in Snord67KO compared to D3H2 WT cells. Enrichment analysis was performed in R using the enrichGO function with a *p*-value cut-off of < 0.05, and the Benjamini-Hochberg procedure was applied to adjust for multiple testing. Individual *p*-values are provided in Supplementary Data 4. **C** Distribution of types of alternative splicing events in Snord67KO compared to D3H2

WT cells. CE cassette exon, RI retained intron, A3SS alternative 3′-splice site, A5SS alternative 5′-splice site, MXE mutually exclusive exons. Splicing diagrams created in Adobe Illustrator. **D** Volcano plots of differential CE events with |ΔPSI| ≥ 0.1 and FDR < 0.05 in Snord67KO cells compared to D3H2 WT cells. Exons with significantly increased inclusion in Snord67KO cells relative to D3H2 WT cells are shown in blue, while exons with significantly decreased inclusion in Snord67KO cells are shown in red. PSI percent spliced in, FDR false discovery rate. **E** Gene ontology analysis of genes with differential CE events in Snord67KO cells versus D3H2 WT cells. Individual *p*-values are provided in Supplementary Data 4.

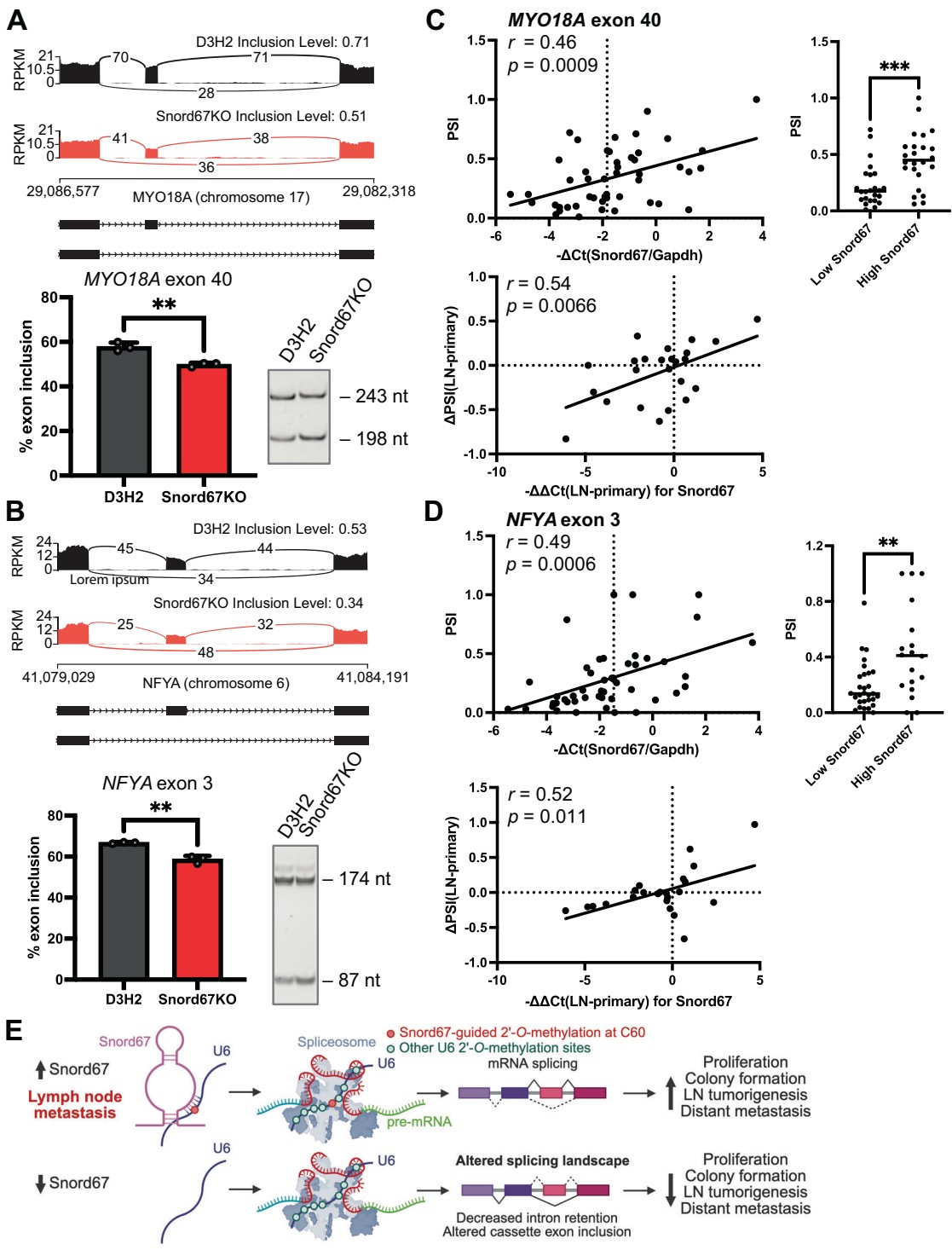

**Fig. 8 | Snord67-dependent differential alternative splicing of *MYO18A* exon 40 and *NFYA* exon 3 in D3H2 cells and breast cancer patients. A, B** Sashimi plots (*top*) and RT–PCR gel with quantification by densitometry (*bottom*) for differentially spliced exon 40 of the *MYO18A* gene (**A**) and exon 3 of the *NFYA* gene (**B**) in Snord67KO cells versus D3H2 WT cells, n = 3 biological replicates (RNA-seq and RT–PCR). Bar plots show mean values, error bars = +1 S.D. Statistical significance was determined by two-tailed t-test; **p < 0.01 (*MYO18A*: p = 0.0028; *NFYA*: p = 0.0016). RPKM, Reads Per Kilobase of transcript per Million mapped reads. **C, D** *Top:* Association between exon inclusion (PSI) and Snord67 expression ($-\Delta C_T$ normalized to GAPDH expression) in breast and lymph node (LN) tumors from breast cancer patients for *MYO18A* exon 40 (**C**) and *NFYA* exon 3 (**D**). n = 48 (**C**) or 47 (**D**) specimens from 24 patients. Linear regression and two-tailed Pearson correlation (*left*, *MYO18A*: p = 0.0009; *NFYA*: p = 0.0006). Two-tailed t-test comparing low and high Snord67 expression groups (*right*, *MYO18A*: p = 0.0005; *NFYA*:

p = 0.0015); **p < 0.01, ***p < 0.001. Dotted line (*left*): threshold for low vs. high Snord67 expression. *Bottom:* Linear regression and two-tailed Pearson correlation between change in exon inclusion (ΔPSI) and change in Snord67 expression ($-\Delta\Delta C_T$ normalized to GAPDH expression) in each LN tumor relative to the matched primary breast tumor from breast cancer patients, for *MYO18A* exon 40 (**C**; p = 0.0066) and *NFYA* exon 3 (**D**; p = 0.011). PSI, percent spliced in; Ct, threshold cycle. **E** Model for effects of Snord67 on U6 methylation, splicing, and LN tumor growth and metastasis. Snord67 levels are increased in LN metastases. In the proposed model, Snord67-guided 2′-*O*-methylation of U6 snRNA at C60 leads to changes in splicing programs, which in turn promote LN tumor growth and distant metastasis. Conversely, loss of Snord67 leads to the observed decrease in U6 C60 methylation, changes in splicing, and decreased LN tumor growth and metastasis. Created in BioRender. Zhou, K. (2025) https://BioRender.com/m73m392. Source data are provided as a Source Data file.

and *CKAP5* would be associated with a greater effect on survival, and we observed that HER2-enriched breast cancer patients with high Snord67 and high *CKAP5* expressing tumors had markedly decreased overall survival compared to patients with low Snord67 and low *CKAP5* expressing tumors (Supplemental Fig. 12G). Together, these results suggest that high Snord67 expression is associated with decreased survival in subsets of patients with aggressive breast cancer subtypes, notably HER2-enriched and Luminal B.

## Discussion

Metastasis is a highly complex, multi-step cascade that requires tremendous phenotypic plasticity in order for cancer cells to survive multiple microenvironments. Metastasis can occur by either hematogenous or lymphatic dissemination, and the lymphatic route was more efficient than the hematogenous route in a microsurgical immune-competent murine model[8,43]. In combination with our prior study examining protein-coding genes that are dynamically regulated within LNs[8], the findings presented here underscore that distinct mechanisms within the unique LN microenvironment can promote metastatic tumor growth. Whereas our previous study focused on protein-coding genes[8], here we focus on non-coding RNAs that are differentially expressed in LN tumors. Interestingly, we found that snoRNAs were overrepresented among ncRNAs that were differentially regulated in LN tumors relative to primary tumors or distant metastases in our murine model (Fig. 1B). Previous studies have shown that snoRNAs, as well as the protein components of small nucleolar ribonucleoproteins (snoRNPs), are overexpressed across multiple tumor types[11,18,44,45]. However, our finding differs from these previous studies in that, rather than demonstrating the global dysregulation of snoRNAs in cancer compared to normal cells, we identify a small subset of snoRNAs that are specifically upregulated in LN metastases compared to other tumor sites. Whereas the global upregulation of snoRNAs in cancer could reflect a broader upregulation of any factors involved in ribosome biogenesis in the setting of rapid cell proliferation, the upregulation of select snoRNAs specifically in LN tumors suggests that this subset of snoRNAs may have unique roles in helping tumor cells survive in the LN microenvironment. Notably, we found that the snoRNAs Snord67 and Snord111 were upregulated both in micro-injected LNs and in de novo LN metastases, raising the possibility that the LN microenvironment induces the upregulation of these snoRNAs, perhaps as an adaptive mechanism in tumor cells.

From our initial screen for ncRNAs that are differentially regulated in LN tumors, we identified the box C/D snoRNA Snord67 as having essential roles for tumor growth and survival in LNs, and perhaps even more importantly, for subsequent metastasis to distant sites. Although there has been increasing appreciation of a role for snoRNAs in cancer[11,20,46,47], few studies have explored the function of snoRNAs in metastasis. While one study found that loss of the box H/ACA snoRNA Snora23 led to decreased tumor growth and metastasis in a xenograft model of pancreatic cancer, the mechanisms by which snoRNAs

regulate metastasis have not been delineated[48]. SnoRNAs canonically function in guiding 2′-*O*-methylation or pseudouridylation of rRNA and snRNA. Here, we provide experimental confirmation that Snord67 is required for 2′-*O*-methylation of its predicted target site, the C60 position of U6 snRNA, in murine and human breast cancer cell lines (Figs. 2C, 3E, and 3H)[30]. Moreover, by introducing a point mutation that prevents Snord67 from guiding 2′-*O*-methylation of U6 at C60, we demonstrate that this methylation event is critical for the function of Snord67 in promoting in vitro colony formation and spheroid growth (Fig. 2D, E). We found that loss of Snord67 leads to changes in the splicing landscape of both murine and human breast cancer cells (Figs. 6, 7), resulting in differential alternative splicing of cassette exons as well as decreased overall intron retention. Based on these results, we suggest a model in which Snord67-guided 2′-*O*-methylation of U6 C60 in the spliceosome is required for baseline splicing patterns, while loss of Snord67 results in loss of U6 2′-*O*-methylation, leading to an altered splicing landscape (Fig. 8E). An impact of 2′-*O*-methylation of U6 snRNA on splicing is supported by prior studies showing that loss of the RNA-binding protein La-related protein 7, which facilitates 2′-*O*-methylation of U6, induces significant changes in mRNA splicing[49,50]. Future studies will be needed to determine the molecular mechanisms by which snoRNA-directed 2′-*O*-methylation of U6 may lead to changes in alternative splicing and overall splicing efficiency. It is possible that 2′-*O*-methylation of U6 affects binding of splicing factors or RNA-binding proteins, or perhaps leads to conformational changes in U6 that alter its interactions with other spliceosome components[22,51–53].

The dysregulation of splicing is known to impact cancer progression and metastasis[54–56]. In fact, mutations in splicing factors often occur in cancer, and recurrent point mutations in snRNAs have been reported in several cancer types[57]. We suggest that snoRNA-guided snRNA modifications, similar to point mutations in snRNAs[57,58], may impact tumor growth and metastasis by modulating splicing efficiency or by promoting alternative splicing. In particular, we propose that the loss of Snord67-guided 2′-*O*-methylation of U6 C60 and the resulting changes in splicing are responsible for the decrease in LN tumor growth and LN-derived distant metastases observed upon Snord67 knockout (Fig. 8E). We found that loss of Snord67 led to overall increased splicing efficiency, as evidenced by decreased intron retention. Intron retention is frequently increased in cancer compared to normal tissues[59] and is a common mechanism for tumor suppressor inactivation in cancer cells[60,61]. In patients with cancer, somatic single-nucleotide variants that cause intron retention are enriched in tumor suppressor genes, where they frequently lead to introduction of a premature termination codon and result in tumor suppressor inactivation[60]. In addition to genetic alterations, protein binding and post-transcriptional modification can target tumor suppressor transcripts for intron retention and nuclear decay, thereby promoting cancer growth[61]. Here, we suggest that Snord67-guided 2′-*O*-methylation of a core spliceosomal component, U6 snRNA, facilitates intron

retention. This global effect on intron retention may contribute to cancer cell survival and egress from lymph nodes, either through tumor suppressor inactivation or through other mechanisms. Besides affecting overall splicing efficiency, Snord67 may also promote LN metastasis by modulating the alternative splicing of specific cassette exons. We found that exon 40 in the *MYO18A* gene and exon 3 in the *NFYA* gene were not only downregulated in Snord67 knockout cells, but also positively correlated with Snord67 expression levels in breast cancer patients. Notably, these two alternatively spliced exons have previously reported roles in breast cancer. Inclusion of exon 40 in *MYO18A* has been reported to promote a mesenchymal phenotype in breast cancer cells and is associated with poor prognosis in patients with triple-negative breast cancer[39]. Inclusion of exon 3 in *NFYA* has been shown to promote tumor growth in murine models[62] and is associated with poor prognosis in patients with basal-like breast cancer[40]. Thus, the Snord67-dependent inclusion of these two exons could contribute to the decreased LN metastasis of Snord67 knockout cells observed in our murine models. In patients with breast cancer, increased Snord67 expression may similarly lead to decreased splicing efficiency and alternative splicing, resulting in more aggressive tumor phenotypes. Although Snord67 expression was not increased in the LN relative to the primary breast tumor for all patients, the upregulation of Snord67 may be one of several possible mechanisms that promote lymphatic metastasis. In a subset of patients, the upregulation of Snord67 in LN metastases may lead to changes in the splicing landscape, which in turn promote LN tumor growth and distant metastasis. Further studies will be needed to elucidate the contribution of specific Snord67-associated splice variants to the Snord67-dependent promotion of breast cancer lymphatic metastasis.

Beyond their canonical function of guiding modifications on rRNAs and snRNAs, snoRNAs have also been found to perform multiple non-canonical functions. For instance, snoRNAs can guide the 2′-O-methylation of non-canonical targets such as mRNAs, and some snoRNAs have RNA modification-independent roles in miRNA-like gene repression and mRNA 3′ processing[13,21,63–65]. Notably, snoRNAs have been proposed to modulate splicing through non-canonical mechanisms, including masking of exonic splicing silencer elements and competition for U1 small nuclear ribonucleoprotein binding sites[13,63,66]. Thus, although Snord67 guides methylation of a core spliceosome component, U6 snRNA, it is also possible that Snord67 modulates splicing through mechanisms independent of U6 methylation. Critically, we showed that a Snord67 construct with a point mutation in the antisense element fails to rescue U6 C60 methylation and only partially rescues in vitro colony formation and spheroid area in Snord67 knockout cells (Fig. 2), suggesting that the canonical function of Snord67 in guiding U6 C60 methylation is important for its role in promoting tumor growth. However, this result does not exclude a role for non-canonical targets of Snord67-guided methylation that also depend on an intact antisense element. Although we did not identify any other targets of Snord67-guided 2′-O-methylation by RiboOxi-seq, this method has limited sensitivity for 2′-O-methylation sites in less abundant RNAs, and it remains possible that Snord67 guides the 2′-O-methylation of RNAs other than U6. Furthermore, the observation that the mutant Snord67 construct can still partly rescue in vitro phenotypes raises the possibility that other, non-canonical mechanisms may contribute to the role of Snord67 in cancer cells. Therefore, Snord67 may modulate splicing and lymphatic metastasis both through its canonical mechanism of guiding U6 methylation and through non-canonical mechanisms, such as the 2′-O-methylation of targets other than U6 or other methylation-independent mechanisms. Further studies will be needed to explore a possible role for such non-canonical mechanisms of Snord67.

Finally, our results support a potential role for ASO technology in therapeutically targeting cancer metastasis, either through the use of ASOs to target dysregulated, pro-metastatic snoRNAs, or potentially through the use of ASOs that are specifically designed to target metastasis-specific alternative splicing events. Of note, splice site-specific ASOs such as nusinersen have demonstrated remarkable clinical efficacy in other applications such as spinal muscular atrophy[67,68]. Future work directed at further elucidating the mechanisms by which snoRNAs modulate splicing and promote metastasis could therefore lead to novel strategies for treating cancer patients.

## Methods

### Study ethics and approval
This study complies with all relevant ethical regulations. All animals were cared for according to guidelines set forth by the American Association for Accreditation of Laboratory Animal Care and the U.S. Public Health Service policy on Human Care and Use of Laboratory Animals. All mouse studies were approved and supervised by the University of North Carolina at Chapel Hill Institutional Animal Care and Use Committee. The maximum tumor size permitted by our approved animal protocol is 2 cm in greatest diameter; this maximum tumor size was not exceeded in this study.

### Sex as a biological variable
Our study exclusively examined female mice, since the vast majority of triple-negative breast cancer occurs in women.

### Cell lines
4T1 cells were obtained from the ATCC (CRL-2539) and maintained in RPMI medium containing 10% v/v FBS and 100 U/mL penicillin/streptomycin. 4T1 cells were transduced with lentiviral constructs expressing either GFP/firefly luciferase or mCherry/Renilla luciferase and then selected and maintained in puromycin (4 μg/mL). EO771.LMB cells were provided by Dr. Robin Anderson and were maintained in DMEM with high (4.5 g/L) D-glucose, 20 mM HEPES, 10% v/v FBS, and 100 U/mL penicillin/streptomycin. MDA-MB-231-D3H2LN-luc (D3H2) cells were provided as a kind gift from Dr. Zdravka Medarova and were maintained in DMEM with high (4.5 g/L) D-glucose, 10% v/v FBS, and 100 U/mL penicillin/streptomycin. All cells were routinely tested for mycoplasma using the Lonza MycoAlert Detection kit (LT07-418).

### Patient samples
We obtained snap-frozen tissue from 24 matched pairs of primary breast tumors and lymph node tumors from patients with breast cancer under a protocol approved by the University of North Carolina at Chapel Hill Institutional Review Board (IRB 21-2233). All subjects were female. Informed consent was obtained from all subjects.

### Generation of CRISPR knockout cell lines
All-in-one CRISPR/Cas9 clones targeting Snord67 and Snord111 were purchased from Genecopoeia (vector pCRISPR-CG02). sgRNA sequences were designed using CRISPOR[69]. 4T1, EO771.LMB, and MDA-MB-231-D3H2LN-luc cells were transiently transfected with pairs of Cas9–sgRNA constructs by electroporation using the Neon Transfection System (Invitrogen). Single cell clones were isolated by serial dilution. To confirm Snord67 gene deletion, DNA was isolated using the DNeasy Blood and Tissue Kit (Qiagen, Cat 69504) and then sequenced (Eton Biosciences). Only single cell clones with confirmed DNA deletion by sequencing and loss of RNA expression by RT−qPCR were used for further phenotypic evaluation. The sequences of the sgRNAs and sequencing primers are provided in Supplementary Data 3.

### Generation of Snord67 overexpression and mutant overexpression cell lines
A custom plasmid of Snord67 flanked by its two exons was created (Genecopoeia) to rescue Snord67 expression (OE) in Snord67

knockout cells. Phusion Site-Directed Mutagenesis kit (Thermo Scientific) was used to direct a single base pair change (G97C) on Snord67 to create a mutant Snord67 overexpression plasmid (mutOE). We then used Sanger sequencing to identify a clone with the desired base mutation. Next, we used NheI (NEB) to linearize OE and mutOE plasmids for stable transfection. Linearized plasmids were transfected into 4T1 Snord67 knockout cells using Lipofectamine 2000 (Thermo Scientific). Cells were grown in 50 μg/mL hygromycin for selection, and then GFP-positive clones were isolated using a fluorescence microscope. Cells were maintained in culture medium containing 50 μg/mL hygromycin, but hygromycin was omitted at the time of plating for colony formation and spheroid formation assays. The sequences of the primers used for site-directed mutagenesis and for Sanger sequencing are provided in Supplementary Data 3.

### Reverse transcription and quantitative PCR (RT−qPCR) and RT−PCR

Total RNA was purified from cell lines using the Zymo Quick-RNA miniprep or microprep kit (Zymo Research, R1057 or R1050). cDNA was synthesized using the SuperScript First-Strand cDNA Synthesis Kit (Thermo Scientific, 11904018) using gene-specific primers for stem-loop qPCR or using the iScript cDNA Synthesis Kit (Bio-Rad, 1708890). For qPCR, cDNA combined with SYBR Green Master Mix (Bio-Rad, 1525271) for measurement on a StepOnePlus qPCR machine (Applied Biosystems). Data was analyzed using the ΔΔCt method and normalized to GAPDH or RPLP0. For measurement of Snord67 levels in clinical samples, qPCR reactions consisted of 6.25 μL 2× PowerUp SYBR Green Master Mix (Applied Biosystems), 2 μL 1:4 diluted cDNA, 0.5 μL 20 μM Forward Primer, 0.5 μL 20 μM Reverse Primer and 3.25 μL water for a total reaction volume of 12.5 μL. The thermal cycling conditions consisted of 50 °C for 2 min, 95 °C for 10 minutes then 40 cycles of 95 °C for 15 s followed by 60 °C for 1 minute. Reactions were carried out using a QuantStudio 6 (Applied Biosystems). For PCR to detect possible splice variants of CKAP5, cDNA was combined with 2.5 units/reaction HotStarTaq DNA polymerase, 200 μM dNTPs, and 250 nM primers in 1x PCR buffer with 1x Q-Solution (Qiagen, 203203), and the thermocycler was set to: (i) 95 °C for 15 min, (ii) 30 cycles of 95 °C for 30 sec, 55 °C for 30 sec, 72 °C for 1 min, (iii) 72 °C for 10 min, (iv) 4 °C hold. The PCR products and a 100 bp DNA ladder (NEB, B7025) were run on a 1.4% agarose gel containing SybrSafe stain in TAE buffer at 100 V for 1 hour. The gel was imaged on a ChemiDoc system (Bio-Rad), and Image Lab software (Bio-Rad) was used to quantify the bands by densitometry. The primers used for RT−qPCR and RT−PCR are listed in Supplementary Data 3.

### Measurement of 2′-O-methylation by Nm-VAQ

Adapted from previously published protocol[31]. 2′-O-methylated RNA/DNA chimeras were designed to target the C60 2′-O-methylated site of U6 snRNA and ordered from Dharmacon Horizon. 200−500 ng of total RNA was mixed with 50 pmol of appropriate chimera in 10 mM Tris pH 7.0 buffer and mixed thoroughly. The samples were denatured at 95 °C for 1 minute and immediately moved to ice. One half of the annealed reaction was mixed with RNase H and 10× RNase H Buffer (New England Biolabs), and the other half was mixed with RNase-free water in place of RNase H. Both samples were incubated at 37 °C for 30 minutes before being stored on ice. RNase H was inactivated by heating at 90 °C for 10 minutes. After denaturation, the samples were diluted 1:50, and 5 μL of the dilution was used for cDNA synthesis using SuperScript III reverse transcriptase (Thermo Scientific, 18080051) and a U6 snRNA gene specific primer, then cDNA was diluted again at 1:50, and 4 μL was used for RT−qPCR using Power SYBR Green PCR Master Mix (Applied Biosystems, 4368577). Alternatively, after treatment with or without RNase H, the samples were cleaned up using the Monarch 10 μg RNA

Cleanup Kit (NEB, T2030) and eluted in 10 μL RNase-free water, and 1 μL of the elution was used for cDNA synthesis using SuperScript III reverse transcriptase and random hexamers, and then 2 μL (one tenth of the reaction volume) was used for qPCR using Power SYBR Green PCR Master Mix. qPCR primers were designed to flank the RNase H cleavage site. Percent methylation at C60 in U6 snRNA was determined by comparing the amplification in the RNase H-treated sample to the amplification in the untreated sample, with normalization to either GAPDH or RPLP0 in the samples both with and without RNase H. Sequences of chimeras and qPCR primers for Nm-VAQ are listed in Supplementary Data 3.

### Protein extraction and Western blotting

Cultured cells were washed with PBS, detached with trypsin, and spun down into cell pellets. Cell pellets were washed with PBS and resuspended in cold RIPA Lysis and Extraction Buffer (Thermo Scientific, 89900) with 1× Halt protease inhibitor (Thermo Scientific, 87786). For wild-type and Snord67 knockout cells, the resuspended cells were combined with ceramic beads (Thermo Scientific, 15-340-154) and lysed in with a bead mill in three 30-second pulses at 1 meter/second at 4 °C, incubating for 2 min on ice between each 30 s pulse. The lysates and beads were then centrifuged at 10,000 × g at 4 °C for 1 min, and the supernatant was transferred to a fresh tube. The lysates were rotated at 4 °C for 20−40 minutes, vortexed for 10 s, incubated at room temperature for 5 min, loaded on a QIAshredder column (Qiagen, 79656), and centrifuged at 21,000 × g for 2 min at room temperature, and the flow-through lysate was transferred to a new tube. For ASO-treated cells, cells resuspended in RIPA buffer with protease inhibitor were incubated on ice for 20 minutes with intermittent mixing, and then centrifuged at 14,000 × g for 15 minutes at 4 °C, and the supernatant lysate was transferred to a new tube. The protein concentration was measured using the Qubit Protein Assay Kit (Thermo Scientific, Q33212). For Western blotting, 20−50 μg of protein was combined with 4× Laemmli sample buffer (Bio-Rad, 1610747) with 10% v/v β-mercaptoethanol and heated at 95 °C for 5 min. The protein samples, MagicMark XP Western Protein Standard (Thermo Scientific, LC5602), and Precision Plus Dual Color Standards (Bio-Rad. 1610374) were run on a 7.5% Mini-PROTEAN TGX precast gel (Bio-Rad, 4561026 or 4561023) in Tris/Glycine/SDS running buffer (Bio-Rad, 1610732) at 200 V for 30 minutes, and then transferred to a polyvinylidene fluoride membrane (Millipore, IPVH00010) at 100 V for 2 h at 4 °C. The membranes were blocked in 5% w/v milk (Bio-Rad, 170-6404), 20 mM Tris-Cl (pH 7.6), 150 mM NaCl, and 0.1% v/v Tween20, and then probed with 1/1000 rabbit anti-CKAP5 polyclonal antibody (Thermo Scientific, PA3-16835) or 1/400 mouse anti-vinculin monoclonal antibody (Sigma, V9131) at 4 °C overnight, followed by 1/2000 HRP-linked horse anti-rabbit IgG (Cell Signaling Technology, 7074P2) or 1/2000 HRP-linked horse anti-mouse IgG (Cell Signaling Technology, 7076P2). Blots were imaged on a ChemiDoc system (Bio-Rad), and Image Lab software (Bio-Rad) was used to quantify the bands by densitometry.

### Synthetic snoRNA transfection

Fully synthetic human Snord75 (control) snoRNA was chemically synthesized (Dharmacon, Inc.), and murine Snord67 was in vitro transcribed using the T7 RiboMAX Express Large Scale RNA Production System (Promega P1320). The RNA was reconstituted at 20 μM in sterile, nuclease-free water. For reverse transfection, 3.125 μL snoRNA or water (negative control) was complexed with 6.25 μL transfection reagent (Lipofectamine RNAiMax, Invitrogen, 2:1 ratio) in 1.25 mL Opti-MEM I medium (Gibco). 500 μL transfection medium was mixed with 2 mL complete medium containing 400,000 EO771.LMB WT or Snord67 knockout cells for a final transfected snoRNA concentration of 10 nM. Reverse transfections were performed in 6-well tissue culture plates (Corning), and transfection medium was exchanged for complete medium after 6 h. At 24 hours post transfection, cells were lifted

and seeded at densities of 100k, 75k, 50k, 25k, 10k, and 5k for time-points of 48, 72, 96, 120, 144, and 168 h post transfection, respectively. Excess cells from the seeding were saved for RNA isolation as the 24 h post transfection timepoint. The sequences of the synthetic snoRNAs are listed in Supplementary Data 3.

## Migration assay

A total of 25,000 4T1 cells were added in serum-free medium to Bio-Coat Control Cell Culture Inserts with 8 μm pores (Corning, 354578) that were pre-coated with 10 μg/mL type 1 rat tail collagen on the bottom of the inserts. RPMI medium containing 10% FBS was added to the lower chambers as the chemoattractant. After 18 h, cells that had migrated to the bottom of the filter were fixed and stained using the Protocol Hema 3 staining kit (Fisher Scientific, 22122911). Membranes were mounted onto glass slides, and images were taken using a Nikon microscope. Migrated cells were enumerated using CellProfiler open source image analysis software.

## Proliferation assay

Cells were plated at a density of 2500 cells per well in a 96-well plate in 120 μL of RPMI containing 10% fetal bovine serum. The plate was placed in an IncuCyte ZOOM Live-Cell Imaging system (Essen Bioscience) (10× objective) and the percent confluence was recorded every two hours by both phase contrast and fluorescence scanning for 96 h at 37 °C and 5% $CO_2$. Images were analyzed using the IncuCyte ZOOM software and the percentages of cell confluence were calculated over time.

## Colony formation assay

After treatment, cells were trypsinized, counted, and 1000 or 5000 cells were plated in triplicate in six-well plates containing complete medium. For the synthetic snoRNA rescue experiment, cells were transfected as described above. 24 h post transfection, cells were washed, trypsinized, and 4000–5000 cells were seeded in 6-well plates in complete medium. Cells were allowed to grow under standard conditions for at least 3 days until colonies were observed. For staining, 1 mL of crystal violet stain (0.05% crystal violet, 1% formalin, 1% methanol in PBS) was added to the cells. Images were taken using an Epson office scanner under film settings. Colonies were quantified using the ColonyArea plugin on ImageJ[70].

## Spheroid assay

Cells were washed, trypsinized, and resuspended in complete medium to a concentration $10^6$ cells/mL, and then 5 μL of the resuspended cells were mixed with 45 μL Matrigel basement membrane matrix (Corning, 356237) and plated in a 24-well plate (5000 cells per well). The 24-well plate was placed at 37 °C, 5% $CO_2$ for 30 min to solidify, and then 750 μL complete medium was added. Brightfield images were taken at 3 days (4T1) or 7 days (D3H2) after plating on a Zeiss Axio Observer Z1/7 inverted microscope with an EC Plan-Neofluar 5x/0.16 Ph 1 M27 objective and an Axiocam 712 mono camera, using Zen 3.6 blue edition version 3.6.095.08000 software. Spheroid images were analyzed using OrganoSeg software.

## Animal studies

For all in vivo studies, adult female Balb/c or C57BL/6 mice (*Mus musculus*) with age 6–8 weeks were purchased from Taconic Farms or The Jackson Laboratory and randomized into experimental groups. The mice were housed on a 12 h light cycle from 7 am to 7 pm, at an ambient temperature range of 68–74 degrees Fahrenheit with a humidity range of 30–70%.

## Mammary fat pad injection

Cells were trypsinized and suspended in Matrigel at a 1:1 ratio, and 50,000 cells were injected directly into the 8th mammary fat pad of mice. Caliper measurements of subcutaneous tumor growth were taken twice weekly, and the tumor volume was calculated as $L \times W^2$ where L is the greatest cross-sectional length across the tumor and W is the length perpendicular to L.

## Axillary lymph node injection

Mice were anesthetized, depilated, and subjected to surgical implantation of 5000 4T1 cells in a total volume of 1 μL Hank's Buffered Saline Solution (HBSS). Injections were performed using a dissecting microscope, a 10 μL Hamilton syringe, and a custom-made microtip Pasteur pipette. Caliper measurements of tumor growth were taken twice weekly, and the tumor volume was calculated as above.

## Tail vein injection

Mice were injected with $1 \times 10^5$ cells in HBSS by tail vein, after which mice were monitored daily for health. Mice were sacrificed and analyzed at the indicated time point post-injection.

## ASO treatment

Second-generation ASOs with a phosphorothioate backbone with 10 central DNA bases flanked by five 2′-methoxyethyl (MOE) modified bases at the 5′ and 3′ ends were designed to target the antisense element of Snord67 and synthesized by Integrated DNA Technologies. For in vitro treatment of 4T1 cells with ASO, negative control or Snord67 ASO was added to the culture medium at a final concentration of 1 μM. For in vitro transfection of D3H2 cells with ASO, negative control or Snord67 ASO was complexed with transfection reagent (Lipofectamine RNAiMax, Invitrogen, 2:1 ratio) in Opti-MEM I medium (Gibco). ASO-containing medium was mixed with 2 mL antibiotic-free medium containing 400,000 D3H2 WT or Snord67 knockout cells for a final transfected ASO concentration of 10 or 40 nM. Reverse transfections were performed in 6-well tissue culture plates (Corning), and transfection medium was exchanged for complete medium after 24 h. At 24 h post transfection, cells were lifted and seeded for spheroid assays and for RNA collection time points. For in vivo ASO experiments, mice were micro-injected with 4T1 cancer cells in the AxLN as described above. Two weeks after injection of cancer cells into the axillary lymph node, PBS or ASO treatment was initiated. ASO was resuspended in 200 μL PBS and injected subcutaneously in the scapular area daily for 6 total treatments.

## Ex vivo cell line establishment

Tumors were excised in a sterile fashion and minced in digestion medium, and tissue was digested for 1 hour in 0.125% collagenase II, 0.1% hyaluronidase, 15 U/mL DNase, and 2.5 U/mL Dispase. Cells were then pelleted, subjected to ACK red blood cell lysis, and plated in 10 cm dishes containing complete RPMI medium and antibiotics as appropriate. For passaging and subsequent culture, ex vivo 4T1 subclones were selected with 6-thioguanine for several days until pure colonies were observed.

## Quantification of lung metastases from AxLN tumors

Lungs were extracted and placed into dissociation buffer that was composed of 1 mg/mL collagenase type 2 (Worthington, #LS004177), 0.25 U/ml neutral protease (Worthington, LS02104), 4 μg/ml deoxyribonuclease (Worthington, #LS006343), and 500 U/ml hyaluronidase (MP Biomedicals, #ICN10074090) resuspended in low-glucose DMEM (Gibco). Tissues were then mechanically minced using the gentleMACS Octo Dissociator (Miltenyi Biotec) and then enzymatically digested at 37 °C for 1 h. Red blood cells were lysed with ACK Lysis buffer, and then cells were plated following several washes with PBS. Cells were cultured at 37 °C for 1 week in complete growth media supplemented with 6-thioguanine. Renilla luciferase activity was then detected by the Renilla Luciferase Assay System (Promega, E2810) using a luminometer.

## Quantification of lymph node and lung metastases from MFP tumors

First, a standard curve was generated by extracting right lung and axillary lymph node from Balb/c or C57bl/6 mice into Trizol reagent (Invitrogen, #15596018). Tissues were dissociated into a cell suspension using a homogenizer. Next, the lung and lymph node cell suspensions were admixed with a known number of 4T1 or EO771.LMB cells, ranging from $10^2$ to $10^6$ cells. Both cell lines were previously transduced with the mCherry reporter gene. RNA purification and cDNA synthesis was performed as described above. mCherry was amplified by qPCR using gene-specific primers (Supplementary Data 3), and the amplicons were resolved in 2% agarose gel followed by quantification using ImageJ software. The standard curve was generated by plotting band intensity versus $\log_{10}$(number of 4T1 or EO771.LMB cells). For the quantification of LN and lung metastases from MFP tumors, four weeks after MFP injection, right lung and axillary LNs were dissected from mice and then underwent RNA extraction, cDNA synthesis, and qPCR for detection of mCherry expression. Amplicons were resolved by gel, and then band intensity was measured quantified by ImageJ. The number of 4T1 and EO771.LMB cells detected in LN and lung tissues was extrapolated using the regression equation obtained from the standard curve.

## Microarray analyses

These microarray results are based on samples belonging to a larger experiment that we performed; some findings coming from this experiment have been previously published[8]. Their annotation and raw files have been deposited in the Gene Expression Omnibus (GEO) repository, with the following accession code: GSE136031. Because those samples have already been described in that article and its GEO metafile, this paragraph of bioinformatics methods aims to 1) summarize what can be found in the aforementioned publication that is relevant also for this new research, 2) highlight differences in the data processing and analysis between that study and this one, and 3) document how the three heat maps that are shown in the main text and supplemental data were generated.

The only samples that are described in the GEO metafile of GSE136031 and were used for the data processing and analyses performed here are: 1) two baseline sample types (4T1-GFP-fLuc and 4T1-mCherry-rLuc), each as a quadruplicate, for a total of 8 samples; 2) two mammary fat pad (MFP) sample types, injected in and harvested from MFP, each as a triplicate (6 samples); 3) three axillary lymph node (AxLN) sample types, injected in and harvested from AxLN, each as a triplicate (9 samples); 4) three AxLN-derived lung metastases (AxLN-LuM), injected in AxLN and harvested from lung metastases (LuM), each as a triplicate (9 samples). Samples were obtained at different time points (see the previously mentioned GEO metafile). The generation and normalization of the RNA expression values relied on the robust multi-array algorithm (RMA)[71,72]. The expression measures of these two plates were combined through a joint scaling procedure (all 41,345 Affymetrix probes combined). However, for this research, we analyzed only those probes that were not included in our previous analysis, which had focused on protein-coding genes that did not have a missing mRNA description and were not poorly characterized in the Affymetrix annotation. These Affymetrix probes formed the collection of heterogeneous and noncoding RNAs used in this analysis. Notably, while this study focuses on a different subset of Affymetrix probes, the included samples are the same as in our previously published article[8].

The analysis started with a tailored differential gene expression (vs. the two baselines) including genes: i) having a mean expression value for at least one sample triplet (*i.e.*, a sample type) ≥ 50% or ≤ 50% than the mean in the two baselines, ii) ≥ 25th percentile in terms of range (across the samples), calculated on the sub-matrix whose columns are the baselines and that sample triplet, and iii) for which every comparison between the three samples of that triplet and the baseline replicates followed either the mathematical relation of being > or of being <. Thereafter, different sample types were compared, following this scheme: a) MFP vs. AxLN, b) MFP vs. AxLN-LuM, and c) AxLN vs. AxLN-LuM. These three comparisons had to fulfill the requirement defined in the above point iii). For the purposes of this research, the two pairwise comparisons that are shown are MFP vs. AxLN and AxLN vs. AxLN-LuM. Therefore, we determined patterns of gene expression across the three samples types (MFP, AxLN, and AxLN-LuM); this analysis was used to identify genes that were differentially expressed in AxLN relative to the set of MFP, AxLN, and AxLN-LuM. For an overview of how the probes for the heat map that displays MFP vs. AxLN vs. AxLN-LuM (24 samples total) were assembled, refer to Leslie et al, since the method used is the same[8].

Finally, we hierarchically clustered the matrix rows (each containing the gene expression values of an Affymetrix ID across the selected samples) on data that were $\log_2$-transformed and mean-subtracted using publicly available software[73,74]. The enrichment of snoRNAs among differentially expressed ncRNAs was evaluated by the hypergeometric test using R Statistical Software version 4.4.0[75].

## Preparation of libraries for RibOxi-seq and RNA-seq of 4T1 cells

Total RNA was isolated from 4T1 cells using the Purelink RNA Mini Kit (Invitrogen). Three biologic replicates were used for each sample. RNA samples were assessed using bioanalyzer to ensure sample quality. mRNA was then enriched using NEBNext Poly(A) mRNA Magnetic Isolation Module. Enrichment was assessed by examining rRNA depletion using a bioanalyzer. For mRNA-seq library prep, the KAPA Stranded mRNA-seq kit (Roche) was used. The methods for the RibOxi-seq library preparation were modified from a previously published protocol[36,37]. In brief, Poly(A) enriched RNA was denatured for 3 min at 90 °C and fragmented by Benzonase (final RNA concentration: 100 ng/µL, enzyme 0.5 units/100 µL reaction) on ice for 90 minutes. Following extraction with acidic phenol-chloroform, the fragmented samples underwent 2 rounds oxidation-elimination cycles. The oxidized RNA was then ligated to the NEB Universal miRNA cloning linker using T4 RNA ligase 2, truncated KQ (NEB, M0373S) at 16 °C overnight. 25 µM RibOxi RT primer was then annealed to the reaction before ligation of 5′ RNA linker using T4 RNA ligase I at room temperature for 1 hour and subsequent reverse-transcription using Protoscript II (NE Biolabs, M0368S). cDNA was then purified with AmpureXP beads with 1:1.8 ratio. 10 µM Illumina compatible forward and reverse PCR primers were added together with Q5 master mix (NE Biolabs, M0544S) followed by AmpureXP purification with 1:1 ratio. A custom cycling program was used to amplify the final PE libraries where 5 cycles were with annealing temperature of 55 °C, while the remaining 23 cycles had annealing temperature of 62 °C. Library QC was done with Agilent bioanalyzer 2200 with DNA1000 chips and reagents. Samples were then sequenced by the UNC High-throughput Sequencing Facility using a HiSeq 4000 (Illumina). Sequences of oligonucleotides used in RNA-seq and RibOxi-seq are provided in Supplemental Table 2.

## Preparation of libraries for RNA-seq of MDA-MB-231-D3H2LN-luc cells

Three biological replicates of MDA-MB-231-D3H2LN-luc WT or Snord67 knockout cells were collected and stored at −80 °C. RNA was extracted from the frozen cell pellets using the RNeasy kit (Qiagen). Libraries were prepared from 1 µg of total RNA for each sample using the KAPA mRNA HyperPrep Kit (Roche) and Illumina adapters (NEB). Agarose gel electrophoresis was used to confirm library size, and Qubit was used to measure the concentration. Pooled libraries were then sequenced for paired end reads on an Illumina NovaSeq X Plus PE150 sequencer.

### Preparation of libraries for RNA-seq of paired breast tumor and lymph node metastases from breast cancer patients

Total RNA was extracted from 10 mg of snap-frozen tissue from paired breast tumors and lymph node metastases from 24 breast cancer patients on a KingFisher Flex instrument (Thermo Scientific, 5400610) using the MagMAX mirVana total RNA isolation kit (Applied Biosystems, A27828). Nucleic acid quantification was performed on a Qubit Flex Fluorometer (Thermo Scientific, Q33327) using the Qubit RNA broad range assay (Invitrogen, Q10211), and RNA quality was evaluated on a TapeStation 4200 instrument (Agilent, G2991AA) by RNA ScreenTape Analysis (Agilent, 5067-5576). Libraries were prepared from 1 μg of total RNA on a Hamilton NGS STAR instrument using the TruSeq Stranded Total RNA Library Prep Gold Kit (Illumina, 20020599). Library quantification was performed on a Qubit 3.0 Fluorometer (Thermo Scientific, Q33216) using the Qubit 1X dsDNA broad range assay (Invitrogen, Q33266). Library quality was evaluated on a TapeStation 4200 instrument (Agilent, G2991AA) by DNA ScreenTape Analysis (Agilent, 5067-5582). The libraries were sequenced on an Illumina NovaSeq 6000 for 2 × 50 base-pair paired end reads at 90 million clusters per library. RNA preparation, library construction, quality control, and sequencing were performed in the University of North Carolina at Chapel Hill Lineberger Comprehensive Cancer Center Translational Genomics Lab (RRID:SCR_025231).

### Analysis of RNA sequencing data from 4T1 cells, MDA-MB-231-D3H2LN-luc cells, and paired breast tumor and lymph node metastases from breast cancer patients

The quality of the raw RNA-seq reads was assessed using FastQC (v.0.12.0)[76] and MultiQC (v1.23)[77]. The RNA-seq reads from the 4T1 cells were mapped to the mouse genome build m39 from Ensembl[78] using STAR (v2.7.11b)[79] to generate bam files. The RNA-seq reads from the D3H2 cells were trimmed using fastp, and then mapped to the human genome build hg38 from Ensembl[78] using STAR[79] to generate bam files. The RNA-seq reads from the paired breast tumor and lymph node metastases from patients were mapped to the human genome build hg38 from Ensembl[78] using STAR[79] to generate bam files. SAMtools (v1.9a)[80] was used to sort the bam files. The mapped reads were further analyzed with rMATS[81] to measure alternative splicing. Significant events were defined as alternative splicing events with ΔPSI ≥ 0.10 and FDR < 0.05, with either total inclusion or exclusion read counts ≥ 5. Transcript quantification was performed using Salmon (v1.10.2)[82] to calculate the level of gene expression. DESeq2 (v3.19)[83] was then used to identify significant genes with a $|log_2(foldchange)| > 1$ and an adjusted $p$-value < 0.05. Gene Ontology (GO) analysis was performed using the R/Bioconductor clusterProfiler function enrichGO[84]. For the overlap analysis, overlapping genes in a background set of only co-detected events were used. The hypergeometric test was used to calculate the significance of the overlap. Splicing efficiency analysis was performed using IRFinder version 2.0[85]. The bam files generated by STAR were used as the input. Only events that were considered 'pure introns' were included. The intron retention ratio was defined as: Intron Retention Ratio = (number of reads with intron included)/(total number of reads). Splicing efficiency was defined as: Splicing Efficiency = 1 − (mean Intron Retention Ratio).

### Analysis of RiboOxi-seq

For RiboOxi-seq analysis, *cutadapt* was used to remove read-through adapters since insert sizes are predicted to be small. *pear* was then used to merge each pair of reads into a single read with default parameters. The actual 3′-linker sequence was removed using *cutadapt*. Mapping was done against the index generated in the mRNA-seq analysis step or U6 snRNA/rRNA transcript sequences. The read 3′-end counts were then generated by processing BED files converted from alignment BAMs. The processed and raw files have been deposited in GEO repository, with accession code GSE269267. For the plots in Supplemental Fig. 11, all positions with 0 reads were converted to 1, and the number of reads at each base position $n$ was normalized to the average number of reads at the neighboring base positions $n-1$ and $n+1$ to generate the "Nm score."

### Validation of alternative splicing by RT−PCR

Total RNA (2 μg) was diluted in nuclease-free water to a final concentration of 200 ng/μL. The High-Capacity cDNA Reverse Transcription Kit (Applied Biosystems, #4368813) was used according to the manufacturer's protocol to reverse transcribe the RNA into cDNA. The thermocycler reaction settings were the following: (i) 25 °C for 10 min, (ii) 37 °C for 120 min, (iii) 85 °C for 5 min, (iv) 4 °C pause. Primers were designed in the upstream and downstream constitutive exons most proximal to the alternative region. The PCR reaction was prepared using GoTaq Green Master Mix (Promega, M7123) and primers diluted to a final concentration of 0.5 μM. The thermocycler reaction settings were the following: (i) 95 °C for 75 sec, (ii) 27 cycles of 95 °C for 45 s, 57 °C for 45 s, 72 °C for 1 min, (iii) 72 °C for 10 min, (iv) 25 °C pause. The PCR products were run on a 6% polyacrylamide gel in TAE buffer (40 mM Tris, 20 mM acetic acid, 1 mM EDTA) at 140 to 150 V for 3 hours, except for validation of the Fn1 alternative splicing event in MDA-MB-231-D3H2LN-luc cells, in which the PCR products were run on a 2% agarose gel at 110 V for 70 minutes. The pUC19 ladder (Thermo Scientific, SM0221) was loaded alongside samples. The gel was submerged in an aqueous solution of 0.4 μg/mL ethidium bromide (MP Biomedicals, ETBC1001) for 10 minutes to stain the DNA. The ChemiDoc XRS+ Imaging System (Bio-Rad) was used to visualize bands, and the Image Lab software (Bio-Rad) was used to quantify the bands by densitometry. The percent spliced in (PSI) values were calculated as reported previously[86]. Sequences of the primers used for RT−PCR are provided in Supplementary Data 3.

### Alternative Splicing Visualization

The rMATS data was visualized using the rmats2sashimiplot program (v3.0.0) (https://github.com/Xinglab/rmats2sashimiplot). BAM and rMATS output files were used to run the program, and replicates were averaged with a group file. Introns were scaled by a factor of 5 for plotting.

### Analysis of correlation between alternative splicing and Snord67 expression in breast cancer patients

For the first correlation analysis, Pearson's correlation was calculated between (1) the PSI for the exon of interest (in *MYO18A* or *NFYA*) as determined from RNA-seq and (2) the $-\Delta C_T$ value for Snord67 expression normalized to GAPDH expression as determined by RT−qPCR, *i.e.* − $C_T$(Snord67) − [−$C_T$(Snord67)]. For the second correlation analysis, Pearson's correlation was calculated between (1) the ΔPSI for the change in PSI of exon of interest (in *MYO18A* or *NFYA*) in the LN metastasis relative to the PSI in the matched primary breast tumor, *i.e.* PSI(LN) − PSI(primary), and (2) the $-\Delta\Delta C_T$ value for the difference in Snord67 expression (normalized to GAPDH) in the LN metastasis compared to the matched primary breast tumor, *i.e.* −$\Delta C_T$(LN) − [−$\Delta C_T$(primary)]. The difference in PSI between the low Snord67 and high Snord67 expression groups was evaluated by t-test, where the threshold Snord67 expression level that yielded the lowest $p$-value was used.

### Analysis of TCGA BRCA data

The Cancer Genome Atlas (TCGA) Breast Invasive Carcinoma (BRCA) data were obtained from the BRCA TCGA Pan-Cancer Atlas 2018[87,88]. SnoRNA expression data were obtained from a previously published analysis of TCGA microRNA sequencing data[11]. Pearson correlation was used to determine the relationship between *CKAP5* and Snord67 expression. The log-rank test was used to evaluate the effect of *CKAP5*

or Snord67 expression on overall survival in patients with specific breast cancer subtypes.

## Statistics

For in vivo experiments, between 5 and 15 mice were assigned per treatment group; this sample size gave approximately 80% power to detect a 50% change in tumor weight with 95% confidence. Results of in vitro and in vivo experiments were compared using Student's *t*-test for comparisons of two groups, and analysis of variance (ANOVA) for multiple group comparisons. For values that were not normally distributed (as determined by the Kolmogorov–Smirnov test), the Mann–Whitney rank sum test was used. A *p*-value less than 0.05 was deemed statistically significant. All other statistical tests for in vitro experiments were performed using GraphPad Prism 8 (GraphPad Software, Inc., San Diego, CA). The multiple hypothesis testing correction of these results was made using the false discovery rate (FDR) method.

## Reporting summary

Further information on research design is available in the Nature Portfolio Reporting Summary linked to this article.

## Data availability

The microarray data used in this study are available in the Gene Expression Omnibus (GEO) data bank under accession code GSE136031 [https://www.ncbi.nlm.nih.gov/geo/query/acc.cgi?acc=GSE136031]. The processed microarray data are provided in the Supplementary Data 1 and Supplementary Data 2 files. The RibOxi-seq data generated in this study have been deposited in the GEO database under accession code GSE269267 [https://www.ncbi.nlm.nih.gov/geo/query/acc.cgi?acc=GSE269267]. The RNA sequencing data for the 4T1 and D3H2 cells generated in this study have been deposited in the GEO database under accession code GSE274590 [https://www.ncbi.nlm.nih.gov/geo/query/acc.cgi?acc=GSE274590]. The RNA sequencing data for the paired primary breast tumors and matched lymph node metastases generated in this study have been deposited in the GEO database under accession code GSE274815 [https://www.ncbi.nlm.nih.gov/geo/query/acc.cgi?acc=GSE274815]. The remaining data are available within the Article, Supplementary Information, or Source Data file. Source data are provided with this paper.

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

## Acknowledgements

The authors acknowledge members of the Pecot lab for helpful discussions and feedback. We also acknowledge the University of North Carolina at Chapel Hill Lineberger Comprehensive Cancer Center Office of Genomics Research (OGR) and Translational Genomics Lab (TGL), who performed the sample processing, library preparation, and RNA sequencing for the clinical samples. Finally, we would like to thank the patients and their families who generously contributed the tumor samples used in this study.

Y.L.C. was supported by the John Pope Fellowship Award from UNC as well as start-up funds from UPMC Hillman Cancer Center and the University of Pittsburgh. K.I.Z. was supported by the Duke Hematology & Transfusion Medicine Training Program (T32 HL007057), the Duke Cancer Institute Cancer Research Young Investigator Pilot Award (part of the Cancer Center Support Grant, P30 CA014236), and the American Society of Clinical Oncology Conquer Cancer Young Investigator Award supported by the Breast Cancer Research Foundation. C.V.P. was supported in part by the National Institutes of Health (NIH) R01CA215075, R01CA258451, R01CA279532, 1R41CA246848, and 1R44CA284932, a Lung Cancer Initiative of North Carolina Innovation and Alumni Award, a Mentored Research Scholar Grant in Applied and Clinical Research (MRSG-14-222-01-RMC) from the American Cancer Society, the Jimmy V Foundation Scholar award, the UCRF Innovator Award, the Stuart Scott V Foundation/Lung Cancer Initiative Award for Clinical Research, the University Cancer Research Fund, the Lung Cancer Research Foundation, the Free to Breathe Metastasis Research Award, the Susan G. Komen Career Catalyst Award, and a NCBC translational research grant. C.L.H., Y.Z., H.A., and E.C. were supported in part by NIH R01HL146381, R01 GM135383, Duke Strong Start Physician-Scientist Award, and the Mandel Foundation. K.K.F. was supported by the UNC Chapel Hill Predoctoral Training Program in Bioinformatics and Computational Biology (5T32GM135123) and the NC Kidney, Urology and Hematology, Training Research, Innovation and Outreach Program (U2CDK133491, TL1DK139567). G.A.G. was supported by the Predoctoral Training Program in Integrative Vascular Biology (5T32HL069768-23). D.D. was supported by NCI R01CA290597 and NIGMS R35GM142864. G.M.G. was supported by NIH-NHLBI 1F31HL170617, and G.M.G. and H.J.W. were both supported by the NIH-NIGMS training award T32 GM119999 and by the NSF Graduate Research Fellowship Program (DGE-1650116). J.G. was supported by start-up funds from the University of North Carolina at Chapel Hill, the National Institutes of Health (NIH R01GM130866 and R35GM152426), a Career Development Award from the American Heart Association (19CDA34660248), and a CAREER Award (#2239056) from the National Science Foundation. The UNC High-Throughput Sequencing Facility is supported in part by an NCI Center Core Support Grant (CA016086) to the UNC Lineberger Comprehensive Cancer Center and the University Cancer Research Fund.

## Author contributions

Y.L.C., K.I.Z., D.D., C.L.H., and C.V.P. were responsible for conceptualization of this project. Y.L.C., K.I.Z., J.G., D.D., C.L.H., and C.V.P. designed the experiments. Y.L.C., K.I.Z., G.M.G., A.C.C., D.J.M., H.L., E.C., L.E., G.A.G., J.J.Z., H.A., H.J.W., L.W., A.E.D.V., and C.V.P. performed the experiments. K.K.F. performed the RNA-seq data analyses. A.P. performed the microarray data annotation, and designed and carried out all microarray analyses. Y.Z. performed the RibOxi-seq analysis. Y.T. and L.A.C. performed data curation for the clinical samples. Y.L.C. and K.I.Z. wrote the paper with input from all authors.

## Competing interests

C.V.P. is the founder of EnFuego Therapeutics, Inc., which is focused on the development of RNA-based therapeutics, and he holds equity in the company. C.L.H. is a co-founder of snoPanTher, which is focused on the development of snoRNA-targeted RNA therapeutics, and he holds equity in the company. The remaining authors disclose no potential conflicts of interest.

## Additional information

[1]University of North Carolina Lineberger Comprehensive Cancer Center, Chapel Hill, NC, USA. [2]Division of Hematology & Oncology, University of North Carolina at Chapel Hill, Chapel Hill, NC, USA. [3]Division of Hematology & Oncology, University of Pittsburgh, Pittsburgh, PA, USA. [4]University of Pittsburgh Medical Center Hillman Cancer Center, Pittsburgh, PA, USA. [5]VA Pittsburgh Health System, Pittsburgh, PA, USA. [6]Division of Medical Oncology, Department of Medicine, Duke University Medical Center, Durham, NC, USA. [7]UNC RNA Discovery Center, University of North Carolina at Chapel Hill, Chapel Hill, NC, USA. [8]Curriculum in Bioinformatics and Computational Biology, University of North Carolina at Chapel Hill, Chapel Hill, NC, USA. [9]Department of Cell Biology & Physiology, University of North Carolina at Chapel Hill, Chapel Hill, NC, USA. [10]Curriculum in Genetics and Molecular Biology (GMB), University of North Carolina at Chapel Hill, Chapel Hill, NC, USA. [11]Division of Cardiology, Department of Medicine, Duke University Medical Center, Durham, NC, USA. [12]Department of Chemistry, University of North Carolina at Chapel Hill, Chapel Hill, NC, USA. [13]Department of Pharmacology, University of North Carolina at Chapel Hill, Chapel Hill, NC, USA. [14]McAllister Heart Institute, University of North Carolina at Chapel Hill, Chapel Hill, NC, USA. [15]Department of Biochemistry and Biophysics, University of North Carolina at Chapel Hill, Chapel Hill, NC, USA. [16]These authors contributed equally: Yvonne L. Chao, Katherine I. Zhou. ✉e-mail: didoming@email.unc.edu; cholley@duke.edu; pecot@email.unc.edu

