## [Transparent Peer Review file · Nature Communications]

Snord67 promotes breast cancer metastasis by guiding U6 modification and modulating the splicing landscape

Corresponding Author: Dr Chad Pecot

Version 0:

Reviewer comments:

Reviewer #1

(Remarks to the Author)

The impact of the study would be significantly improved if their observed findings were more robustly demonstrated in human patients. It is unclear whether SNORD67 is relevant to LN metastasis in patients.

- (1) There is a concern that individual snoRNAs are not driving LN mets since they are more globally dysregulated compared to other ncRNA gene classes (statistics should be added for this analysis).
- (2) What is the mechanism of snoRNA dysregulation more so than other gene classes.
- (3) The selected snoRNAs for subsequent experiments were selected based on their expression within their model, but would be useful to more systematically assess which snoRNAs from their model are also consistently altered in human patients – such as the paired primary and LN mets. This would prioritize candidates that most likely have an impact in patients.
- (4) The data measuring SNORD67 expression is both up- and down-regulated across 28 patients suggests it may not be critical to LN mets.
- (5) Further, the overlap of differential AS between LN mets and primary tumor analysis likely won't reveal much since (i) the expression of SNORD67 in these patients is not provided and (ii) if the expression is both up- and down-regulating in LN mets any subsequent analysis is not factoring SNORD67 levels.
- (6) A more direct association of SNORD67 driven AS should be demonstrated. For instance, if SNORD67 high vs. low expressing patients have similar splicing patterns to their KO models?
- (7) The association of differential AS between LN mets and primary tumors is not necessary occurring at a higher rate than chance. More robust statistics need to be provided to suggest that Snord67 KOs is enriched among the 60 UNC RAP genes. This could be random. "We identified a set of 60 genes with differential AS events between LN metastases and primary tumors that were observed in all six RAP human samples (Figure 7B, left). We evaluated for overlap between the 60 UNC RAP genes and the AS genes discovered in Snord67 knockouts compared to 4T1 WT cells, surmising that these would be most likely to be clinically relevant, and found nine genes in common (Figure 7B, right)."
- (8) If the authors could show more robust evidence that SNORD67 is relevant in human patients, it would also be informative to demonstrate whether the mechanism of methylation observed in their model is also occurring in patients.

Reviewer #2

(Remarks to the Author)

When comparing the transcriptome of cancer cell sub-clones obtained from mammary fat pad tumors, microsurgically-injected axillary lymph node tumors, and spontaneous lung metastases derived from axillary lymph node tumors, the authors noted that affected non-coding RNAs were enriched in snoRNAs, with particularly strong effects for Snord67 and Snord111. They then claim to have specifically inactivated these two snoRNAs without affecting the expression of the host genes (Ckap5 and Sf3b3, respectively), and that clones lacking Snord67 (but not Snord111) display reduced proliferation and colony formation ability. These clones were also shown to display reduce U6 2'-O-methylation (the function of Snord67 is to target U6 for 2'-O-methylation) and to undergo extensive alternative splicing, some of these events affecting genes also undergoing alternative splicing in lymph node metastases from breast cancer patients.

Overall, the manuscript presents potentially interesting findings. Yet, due to the authors' inadequate characterization of the expression of the snoRNA host genes and their testing of only a small number of clones, several of their conclusions lack

robustness.

Figure 2: As well explained by the authors, snoRNA genes are hosted with introns of larger genes, Ckap5 for Snord67 and Sf3b3 for Snord111. Yet, introns also harbor many often cryptic sites regulating splicing, and the authors clearly fail to convincingly demonstrate that the performed genetic engineering does not alter production of the proteins encoded by the host genes. This is an issue as both gene products have been linked to cancer in the literature. The provided data only show qPCRs with primers nested at the 3' ends of the genes, away from the snoRNAs. As a minimum, a Western blot showing that expression of Ckap5 protein is not altered in the clones (in quantity and in size) is absolutely required. RT-PCR reactions with primers nested on Ckap5 exons 30 and 31 (flanking the deletion) would also be reassuring.

Figure 3: To provide evidence for reduced formation of distant metastases upon reduced Snord67 expression, the authors inject only two different Snord67KO clones. Beside the possibility of reduced expression of the snoRNA host gene in these clones, it appears from Figure 4 that the clones are subject to variability (for example, in 4B, Snord67KO-1 yields smaller tumors than WT 4T1 cells, while Snord67KO-2 doesn't). Increasing the number of clones tested would make the conclusions more robust, and the authors should clearly explain how the clones used for the experiments were picked. In this set of experiments, the authors also inhibit Snord67 expression with an antisense oligo. With this ASO, they show reduced tumor volume at high dose and long exposure times. Yet, there is no indication that the ASO may reduce formation of distant metastases. In addition, there seems to be no correlation between the impact of the ASO on Snord67 expression and its effect on tumor volume (3H). The authors mention that this may be a kinetic issue, but this is only speculation. Finally, documenting that the Snord76 ASO does not impact on Ckap5 protein expression would be reassuring.

Figure 5: In Panels D, E, and F (alternative splicing at Eif4a2, Cd44, and Tpm1), the quantification of the splicing events must be performed at least in triplicates to allow for calculation of a standard deviation.

Figure 7: In this figure, the authors document that some genes are affected at the level of splicing in both LN metastases from patients with breast cancer and in mouse Snord67 KO cells. However, from the provided information, it is difficult to assert whether the similarities were at the level of the splicing events or whether they just affected the same genes.

Minor points:

- 1- Supplemental Figure 2D-F on EO771.LMB cells are quoted in the section on 4T1 cells.
- 2- In Figure 5 and 7, the authors use BigWig files to show alternative splicing. This is often misleading, especially when expression levels are modified between conditions. Sashimi plots or their equivalent would be preferable. Systematically drawing the map of the gene in these panels would also facilitate the interpretation.
- 3- Because of the low number of tested clones and validated splicing events, the authors may want to tone down some of their conclusions.

Reviewer #3

(Remarks to the Author)

In the current study, Chao et al. identified that Snord67 contributed to metastases of breast cancer in animal models. The authors demonstrated that Snord67 mediated the 2'-O-methylation of U6, which is a component of the spliceosome. However, the authors only analyzed and described the results of RibOxi-Seq and RNA-seq, lacking to further verify the exact downstream target (CD44, TPM1, and EIF4A2), and did not confirm that the gene is regulated by Snord67-U6 or Snord67, and mediates the function of Snord67. This makes the work around Snord67-U6-metastasis not solid enough. The following issues need to be addressed.

Major,

1. Please upload the AxLN tumors, MFP tumors and AxLN-LuM original microarray data as supplementary table.
2. The phenotypic rescue and Nm-VAQ experiments should be also examined in the 4T1 Snord67 KO cells with Snord67 transfection.
3. Supplemental Figure 4A-B: How about the down-regulation effect of ASO on Snord111? Did the authors design another ASO sequence against SnoRNA?
4. Any photos of the AxLN tumors, lunc metastases tumors after mice sacrifice?
5. The authors verified the 2'-O-methylation of U6 by Snord67 at the beginning of the manuscript (Supplemental Figure 3). Why the RibOxi-Seq was performed to analyze the 2'-O-methylation of U6 later (Figure 5)? Did the authors identify any other 2'-O-methylation target, in addition to U6, by the RibOxi-Seq?
6. The original RibOxi-Seq and RNA-seq data need to be uploaded.
7. Did the authors detect the 2'-O-methylation status of U6 in patients?

Minor,

8. Why the chromatin modifier histone deacetylase 11 was introduced? Any relationship between HDAC11 and Snord67?
9. Move "SnoRNAs are small ncRNAs less than 300 nucleotides...(Figure 1C). ...yet whether and how snoRNAs regulate metastasis is unknown (22-25)." to the introduction section, and supplementary figure.

Version 1:

Reviewer comments:

Reviewer #1

(Remarks to the Author)

There are still some concerns:

- (1) There is still a concern regarding whether it is relevant in human patients.
- (2) SNORD67 expression is both up- and down-regulated across 28 patients suggests it may not be critical to LN mets. While it might be relevant to some patients, a more systematic analysis in patients would have been able to demonstrate whether other RNAs have a more robust and consistent influence on LN mets or if SNORD67 is the optimal candidate.
- (3) The authors have added a correlation between SNORD67 and AS. However, given that SNORD67 can both be up- and down-regulated, it will be important to establish what levels of SNORD67 alter splicing sufficiently to promote LN mets.

Reviewer #2

(Remarks to the Author)

The authors have made substantial improvements to the manuscript, particularly in convincingly demonstrating that the expression of Sno host genes is unaffected by the knockouts and in enhancing the clarity of the figure descriptions.

However, the revision introduces two new figures (Fig 7 and 8) that still require refinement. In these figures, the authors have successfully inactivated Snord67 in human MDA-MB-231-D3H2LN-luc WT cells (D3H2), which represents a valuable new reagent. Despite this, the examples presented in Figures 8A and 8B are not entirely convincing. The observed changes in PSI values are minimal, and the effects are not clearly reflected in the validation PCRs. This may be due to the authors' focus on identifying events that occur both in the cell lines and tumors, and it is probably possible to find better examples without that constrain.

Rather than concentrating on specific alternative splicing events, the authors might consider exploring the potential for reduced splicing efficiency or precision following Snord67 inactivation (or reduced expression). A modified version of Figure 7C, illustrating the number of skipped exons and retained introns under both conditions, could effectively demonstrate reduced splicing activity (as exon skipping and intron retention require fewer splicing events). Additionally, quantifying the total number of split reads and non-annotated splicing events could provide insights into splicing efficiency and precision, respectively (this analysis can be performed using RegTools). Furthermore, calculating the "strength" of de novo splice sites could help identify any reduction in splicing precision.

The goal of these bioinformatics approaches would be to determine whether reduced splicing efficiency, rather than specific splicing events, is observed consistently both in vitro and in tumor samples.

Reviewer #3

(Remarks to the Author)

My concerns have been addressed. Thank you!

Version 2:

Reviewer comments:

Reviewer #1

(Remarks to the Author)

The authors have sufficiently addressed my concerns.

Reviewer #2

(Remarks to the Author)

The claim that Snord67 specifically affects alternative splicing, rather than splicing in general, still does not appear to be strongly supported by the data. Previous comments emphasized the need to examine D3H2 cells inactivated for Snord67 to assess its impact on splicing efficiency. Instead, the authors have focused on 4T1 cells and now provide new data showing a clear overall impact of Snord67 on splicing efficiency. Yet, the conclusions remain centered on "alternative splicing," despite the relatively weak effects observed. In this regard, it should be noted that, as in the previous version, the examples of alternative splicing in Figures 8A and 8B show only minimal changes in PSI values, and these effects are not clearly reflected in the validation PCRs. This issue remains to be addressed.

Main point: A revision of the conclusions, rebalancing the emphasis on splicing versus alternative splicing, is strongly recommended before publication.

Version 3:

Reviewer comments:

Reviewer #2

(Remarks to the Author)

The text has clearly been improved. The paper seems ready for publication.

Reviewer #1: (ncRNA and metastasis)

The impact of the study would be significantly improved if their observed findings were more robustly demonstrated in human patients. It is unclear whether SNORD67 is relevant to LN metastasis in patients.

(1) There is a concern that individual snoRNAs are not driving LN mets since they are more globally dysregulated compared to other ncRNA gene classes (statistics should be added for this analysis).

Although snoRNAs were enriched among ncRNAs identified as being differentially expressed in LN mets, the vast majority of profiled snoRNAs were not differentially expressed, suggesting that this is not simply a matter of global dysregulation of snoRNAs. We have added text to clarify this:

“On the other hand, most snoRNAs (98%) profiled on the microarray were not differentially expressed in AxLN tumors relative to MFP tumors and AxLN-LuM. Thus, a small subset of snoRNAs is responsible for a large fraction of the changes observed in the non-coding transcriptome of AxLN tumors.”

We have added statistics (hypergeometric test) to support that snoRNAs are significantly enriched among ncRNAs that are differentially expressed in LN mets:

“Thus, there was a more than five-fold, statistically significant enrichment of snoRNAs among the differentially expressed ncRNAs ($p = 5.6 \times 10^{-12}$ by the hypergeometric test).”

(2) What is the mechanism of snoRNA dysregulation more so than other gene classes.

While snoRNAs were overrepresented among ncRNAs that were differentially expressed in LN metastases, snoRNAs were not globally dysregulated, so there may not be a common mechanism by which snoRNAs are dysregulated in LN metastases. We have added text to the discussion to make the distinction:

“However, our finding differs from these previous studies in that, rather than demonstrating the global dysregulation of snoRNAs in cancer compared to normal cells, we identify a small subset of snoRNAs that are specifically upregulated in LN metastases compared to other tumor sites. Whereas the global upregulation of snoRNAs in cancer could reflect a broader upregulation of any factors involved in ribosome biogenesis in the setting of rapid cell proliferation, the upregulation of select snoRNAs specifically in LN tumors suggests that this subset of snoRNAs may have unique roles in helping tumor cells survive in the LN microenvironment. Notably, we found that the snoRNAs Snord67 and Snord111 were upregulated both in micro-injected LNs and in de novo LN metastases, raising the possibility that the LN microenvironment induces the upregulation of these snoRNAs, perhaps as an adaptive mechanism in tumor cells.”

(3) The selected snoRNAs for subsequent experiments were selected based on their expression within their model, but would be useful to more systematically assess which snoRNAs from their model are also consistently altered in human patients – such as the paired primary and LN mets. This would prioritize candidates that most likely have an impact in patients.

In general, the measurement of snoRNA levels in patient samples is challenging due to the way RNA-seq is performed (poly(A) selection and size selection). We have added text to the Results section to make this point:

“Since snoRNAs are largely undetected in poly(A)-selected RNA-seq libraries...”

By using a microarray to measure noncoding RNA levels in our murine model of lymphatic metastasis, we were able to systematically assess the changes in expression of snoRNAs and other noncoding RNAs in lymph node metastases relative to primary breast tumors and distant metastases. We then measured Snord67 expression and alternative splicing changes in breast cancer patients to assess whether our findings in mice are likely to also apply to patients.

(4) The data measuring SNORD67 expression is both up- and down-regulated across 28 patients suggests it may not be critical to LN mets.

Although Snord67 expression is not upregulated in the lymph nodes of all breast cancer patients, it may be that Snord67 is important for lymphatic metastasis in a subset of patients. We have added text in the Discussion section to make this point:

“Although Snord67 expression was not increased in the LN relative to the primary breast tumor for all patients, the upregulation of Snord67 may be one of several possible mechanisms that promote lymphatic metastasis. In a subset of patients, the upregulation of Snord67 in the LN tumor may function to promote the alternative splicing of certain exons, such as exon 40 of the *MYO18A* gene and exon 3 of the *NFYA* gene, to promote lymph LN tumor growth and distant metastasis.”

(5) Further, the overlap of differential AS between LN mets and primary tumor analysis likely won't reveal much since (i) the expression of SNORD67 in these patients is not provided and (ii) if the expression is both up- and down-regulating in LN mets any subsequent analysis is not factoring SNORD67 levels.

To determine whether alternative splicing changes observed in patient samples are associated with Snord67 expression levels, we performed an analysis of the correlation between specific alternative splicing events and Snord67 expression in paired primary tumor and LN metastases (Figure 8C–D).

(6) A more direct association of SNORD67 driven AS should be demonstrated. For instance, if SNORD67 high vs. low expressing patients have similar splicing patterns to their KO models?

To demonstrate an association between Snord67 expression and alternative splicing, we performed a correlation analysis. We found that alternative splicing events in *MYO18A* and *NFYA* correlated with Snord67 expression in paired primary breast tumor and LN metastases from breast cancer patients (Figure 8C–D):

“Of the three alternative splicing events validated by RT–PCR in D3H2 cells (Figure 8A–B and Supplemental Figure 12), two events, involving exon 40 of the *MYO18A* gene and exon 3 of the *NFYA* gene, significantly correlated with Snord67 expression in the primary breast and LN tumor specimens from breast cancer patients (*MYO18A* exon 40: Pearson $r = 0.46$, $p = 0.0009$; *NFYA* exon 3: Pearson $r = 0.49$, $p = 0.0006$; Figure 8C–D). Moreover, the change in PSI for these two alternative exons significantly correlated with the change in Snord67 expression in matched pairs of primary breast tumors and LN metastases

(MYO18A exon 40: Pearson $r = 0.54$, $p = 0.0066$; NFYA exon 3: Pearson $r = 0.51$, $p = 0.011$; Figure 8C–D). That is, in patients whose LN metastasis displayed increased Snord67 expression relative to the primary breast tumor, the inclusion of MYO18A exon 40 and NFYA exon 3 was also increased in the LN tumor relative to the primary breast tumor.”

(7) The association of differential AS between LN mets and primary tumors is not necessary occurring at a higher rate than chance. More robust statistics need to be provided to suggest that Snord67 KO is enriched among the 60 UNC RAP genes. This could be random. “We identified a set of 60 genes with differential AS events between LN metastases and primary tumors that were observed in all six RAP human samples (Figure 7B, left). We evaluated for overlap between the 60 UNC RAP genes and the AS genes discovered in Snord67 knockouts compared to 4T1 WT cells, surmising that these would be most likely to be clinically relevant, and found nine genes in common (Figure 7B, right).”

We performed a new analysis using a larger set of 24 paired primary tumor and LN metastases. By performing RNA-seq on these samples, we were able to show that alternative splicing events in *MYO18A* and *NFYA* significantly correlated with Snord67 expression in paired primary breast tumor and LN metastases from breast cancer patients using Pearson correlation analysis (Figure 8C–D).

(8) If the authors could show more robust evidence that SNORD67 is relevant in human patients, it would also be informative to demonstrate whether the mechanism of methylation observed in their model is also occurring in patients.

We were not able to measure U6 methylation in human patients due to limitations in the quantity of isolated RNA. However, to demonstrate that our findings on Snord67 and U6 methylation in murine cell lines also apply to humans, we have shown that Snord67 knockout leads to decreased U6 C60 methylation in the human breast cancer cell line MDA-MB231-D3H2LN (Figure 3H).

Reviewer #2: (Alternative splicing)

When comparing the transcriptome of cancer cell sub-clones obtained from mammary fat pad tumors, microsurgically-injected axillary lymph node tumors, and spontaneous lung metastases derived from axillary lymph node tumors, the authors noted that affected non-coding RNAs were enriched in snoRNAs, with particularly strong effects for Snord67 and Snord111. They then claim to have specifically inactivated these two snoRNAs without affecting the expression of the host genes (*Ckap5* and *Sf3b3*, respectively), and that clones lacking Snord67 (but not Snord111) display reduced proliferation and colony formation ability. These clones were also shown to display reduced U6 2'-O-methylation (the function of Snord67 is to target U6 for 2'-O-methylation) and to undergo extensive alternative splicing, some of these events affecting genes also undergoing alternative splicing in lymph node metastases from breast cancer patients.

Overall, the manuscript presents potentially interesting findings. Yet, due to the authors' inadequate characterization of the expression of the snoRNA host genes and their testing of only a small number of clones, several of their conclusions lack robustness.

Figure 2: As well explained by the authors, snoRNA genes are hosted with introns of larger genes, *Ckap5* for Snord67 and *Sf3b3* for Snord111. Yet, introns also harbor many often cryptic sites regulating splicing, and the authors clearly fail to convincingly demonstrate that the performed genetic engineering does not alter production of the proteins encoded by the host genes. This is

an issue as both gene products have been linked to cancer in the literature. The provided data only show qPCRs with primers nested at the 3' ends of the genes, away from the snoRNAs. As a minimum, a Western blot showing that expression of Ckap5 protein is not altered in the clones (in quantity and in size) is absolutely required. RT-PCR reactions with primers nested on Ckap5 exons 30 and 31 (flanking the deletion) would also be reassuring.

Thank you for raising this important point. We have performed RT-PCR of the CKAP5 transcript using primers flanking the intron that contains Snord67 in the WT and Snord67 KO 4T1, EO771.LMB, and D3H2 lines (Supplemental Figures 2D, 5B, and 6B), and we have performed Western blots showing that CKAP5 protein expression is not significantly altered in the Snord67 KO 4T1, EO771.LMB, and D3H2 cell lines (Supplemental Figures 2E, 5C, and 6C).

Figure 3: To provide evidence for reduced formation of distant metastases upon reduced Snord67 expression, the authors inject only two different Snord67KO clones. Beside the possibility of reduced expression of the snoRNA host gene in these clones, it appears from Figure 4 that the clones are subject to variability (for example, in 4B, Snord67KO-1 yields smaller tumors than WT 4T1 cells, while Snord67KO-2 doesn't). Increasing the number of clones tested would make the conclusions more robust, and the authors should clearly explain how the clones used for the experiments were picked.

Although we tested a limited number of knockout clones in each model, we used multiple different cell line models in our *in vitro* and *in vivo* assays to help support our findings. The Snord67 knockout clones were picked based on confirmation of Snord67 deletion by DNA sequencing and loss of Snord67 expression by RT-qPCR. We have added a sentence in the Methods section to clarify this:

“Only single cell clones with confirmed DNA deletion by sequencing and loss of RNA expression by RT-qPCR were used for further phenotypic evaluation.”

In this set of experiments, the authors also inhibit Snord67 expression with an antisense oligo. With this ASO, they show reduced tumor volume at high dose and long exposure times. Yet, there is no indication that the ASO may reduce formation of distant metastases.

We were not able to assess formation of distant metastases due to the short follow up after injection of the tumor cells in this experiment. We have clarified this point in the text and have added additional text to clarify that the impact of Snord67 loss on distant metastases was only demonstrated with the Snord67 knockout clones and not in the ASO-treated mice:

“Together, these results demonstrate that loss of Snord67 expression, whether by CRISPR knockout or by ASO treatment, leads to decreased AxLN tumor growth in a microsurgical, immune-competent murine model of LN metastasis. Although the effect of Snord67 ASO on distant metastases was not assessed due to the short duration of follow-up, our findings in two Snord67 knockout clones further suggest that loss of Snord67 leads to decreased distant metastases arising from micro-injected AxLN tumors.”

In addition, there seems to be no correlation between the impact of the ASO on Snord67 expression and its effect on tumor volume (3H). The authors mention that this may be a kinetic issue, but this is only speculation.

We have removed the language suggesting that the dose-dependence may be a kinetic issue since this is only speculation.

Finally, documenting that the Snord76 ASO does not impact on Ckap5 protein expression would be reassuring.

We have added Supplemental Figure 6I demonstrating that Snord67 ASO transfection does not affect host protein (CKAP5) expression.

Figure 5: In Panels D, E, and F (alternative splicing at Eif4a2, Cd44, and Tpm1), the quantification of the splicing events must be performed at least in triplicates to allow for calculation of a standard deviation.

We have repeated the analysis of the 4T1 RNA-seq data and validated alternative splicing events by RT-PCR in triplicate (Supplemental Figure 10).

Figure 7: In this figure, the authors document that some genes are affected at the level of splicing in both LN metastases from patients with breast cancer and in mouse Snord67 KO cells. However, from the provided information, it is difficult to assert whether the similarities were at the level of the splicing events or whether they just affected the same genes.

Due to difficulty comparing alternative splicing events in mouse vs. human, as well as poor conservation of alternative splicing between mouse and human, we have now performed RNA sequencing of the human triple negative breast cancer cell line MDA-MB-231-D3H2LN-luc wild-type vs. Snord67 KO (Figure 7). We then compared these alternative splicing events to splicing events observed in paired primary breast tumor and LN metastases from breast cancer patients. We identified specific alternative splicing events (exon 40 in the *MYO18A* gene and exon 3 in the *NFYA* gene) that were differentially spliced in MDA-MB-231-D3H2LN-luc wild-type vs Snord67 KO and also correlated with Snord67 expression in paired primary breast tumor and LN metastases from breast cancer patients (Figure 8).

Minor points:

1- Supplemental Figure 2D-F on EO771.LMB cells are quoted in the section on 4T1 cells.

These figures have been renumbered to accommodate additional data, and we have revised the text to ensure that the relevant figures are quoted.

2- In Figure 5 and 7, the authors use BigWig files to show alternative splicing. This is often misleading, especially when expression levels are modified between conditions. Sashimi plots or their equivalent would be preferable. Systematically drawing the map of the gene in these panels would also facilitate the interpretation.

We have added Sashimi plots to Supplemental Figure 10 showing validation of alternative splicing events.

3- Because of the low number of tested clones and validated splicing events, the authors may want to tone down some of their conclusions.

We have edited our wording in the text to ensure that the stated conclusions are well supported by our data.

Reviewer #3:

In the current study, Chao et al. identified that Snord67 contributed to metastases of breast cancer in animal models. The authors demonstrated that Snord67 mediated the 2'-O-methylation of U6, which is a component of the spliceosome.

However, the authors only analyzed and described the results of RibOxi-Seq and RNA-seq, lacking to further verify the exact downstream target (CD44, TPM1, and EIF4A2), and did not confirm that the gene is regulated by Snord67-U6 or Snord67, and mediates the function of Snord67. This makes the work around Snord67-U6-metastasis not solid enough. The following issues need to be addressed.

Major,

1. Please upload the AxLN tumors, MFP tumors and AxLN-LuM original microarray data as supplementary table.

We have added Supplemental Tables 1–2, which list the ncRNAs that were significantly upregulated or downregulated in AxLN vs MFP tumors and in AxLN-LuM vs AxLN tumors. The original microarray data have been deposited in the GEO repository with accession code GSE136031.

2. The phenotypic rescue and Nm-VAQ experiments should be also examined in the 4T1 Snord67 KO cells with Snord67 transfection.

We have added data showing that stable transfection of 4T1 Snord67 KO cells with a Snord67 expression construct rescues Snord67 levels, U6 C60 methylation, and *in vitro* colony formation and spheroid formation phenotypes (Figure 2).

3. Supplemental Figure 4A-B: How about the down-regulation effect of ASO on Snord111? Did the authors design another ASO sequence against SnoRNA?

We have added data showing that Snord111 is knocked down by ASO (Supplemental Figure 7B). We used a single ASO for each snoRNA target, since ordering a second ASO and using this ASO to repeat these experiments would be very expensive.

4. Any photos of the AxLN tumors, lung metastases tumors after mice sacrifice?

We have added H&E images of micro-injected AxLN tumors and lung metastases derived from micro-injected AxLN tumors (Supplemental Figure 1B–C). Additional H&E images and gross images of our 4T1 microsurgical model have been previously published (Reference 8).

5. The authors verified the 2'-O-methylation of U6 by Snord67 at the beginning of the manuscript (Supplemental Figure 3). Why the RibOxi-Seq was performed to analyze the 2'-O-methylation of U6 later (Figure 5)? Did the authors identify any other 2'-O-methylation target, in addition to U6, by the RibOxi-Seq?

We have edited the text to specify that RibOxi-Seq supports our previous finding that Snord67 guides 2'-O-methylation of U6 snRNA at C60:

“Since Snord67 guides 2'-O-methylation of C60 in U6, we first examined 2'-O-methylation patterns on U6 snRNA in our RibOxi-seq data. In 4T1 WT cells, all eight known 2'-O-methylation sites on U6 snRNA were detected. In Snord67 knockout cells, there was a

loss of methylated reads specifically at site C60, consistent with our Nm-VAQ experiments showing that Snord67 is essential for 2'-O-methylation of C60 in U6 snRNA (Supplemental Figure 11A)."

We also used RibOxi-Seq to explore the possibility that Snord67 may guide 2'-O-methylation at other sites, but we were unable to reliably identify other methylation targets, possibly due to low sensitivity of this method. We have edited the text to clarify this:

"Next, we considered whether RibOxi-seq might reveal non-canonical targets of Snord67-guided 2'-O-methylation. Specifically, since we observed many differential alternative splicing events in Snord67 knockout cells, we wondered whether the differentially spliced pre-mRNAs contained Snord67-guided 2'-O-methylation sites that modulate alternative splicing independent of U6 modification."

"Therefore, we were not able to identify any additional 2'-O-methylation targets of Snord67 besides C60 in U6 snRNA, although we cannot exclude the existence of other Snord67-dependent 2'-O-methylation sites that were not detected at the sensitivity level of RibOxi-seq."

6.The original RibOxi-Seq and RNA-seq data need to be uploaded.

The RibOxi-seq data have been deposited in GEO with accession code GSE269267. The 4T1 and D3H2 RNA-seq data have been deposited in GEO with accession code GSE274590. The RNA-seq of the paired breast tumor and LN metastases from breast cancer patients have been deposited in GEO with accession code GSE274815.

7.Did the authors detect the 2'-O-methylation status of U6 in patients?

We were not able to measure U6 methylation in human patients due to limitations in the quantity of isolated RNA. However, to demonstrate that our findings on Snord67 and U6 methylation in murine cell lines also apply to humans, we have shown that Snord67 knockout leads to decreased U6 C60 methylation in the human breast cancer cell line MDA-MB231-D3H2LN (Figure 3H).

Minor,

8.Why the chromatin modifier histone deacetylase 11 was introduced? Any relationship between HDAC11 and Snord67?

In the previous version of the manuscript, we mentioned HDAC11 to provide context and background, since we previously identified HDAC11 as a protein-coding gene that was differentially expressed in LN metastases. Since HDAC11 does not have direct relevance to the current manuscript, we have removed specific mention of HDAC11 from the text.

9.Move "SnoRNAs are small ncRNAs less than 300 nucleotides...(Figure 1C). ...yet whether and how snoRNAs regulate metastasis is unknown (22-25)." to the introduction section, and supplementary figure.

We have moved the specified text to the introduction and Figure 1C to the supplement (as Supplemental Figure 1A) as recommended.

Dear Colleagues:

We thank the reviewers for their comments. We have addressed the concerns raised by the reviewers. Changes from the previously revised version are denoted in red text in the updated version of the manuscript. Point-by-point responses are in blue text and in figures below.

Reviewer #1 (Remarks to the Author):

There are still some concerns:

(1) There is still a concern regarding whether it is relevant in human patients.

To further investigate the relevance of Snord67 in human patients, we integrated Snord67 expression data from the SNORic database and assessed for associations with survival by PAM50 subtypes.

- We found that Snord67 expression was significantly higher in luminal B and basal subtypes, which are the more clinically aggressive subtypes of breast cancer (panel a, new Supplemental Figure 12B in the manuscript).
- In addition, we found that high Snord67 expression was associated with significantly decreased overall survival in the luminal B subtype (panel b, new Supplemental Figure 12C in the manuscript).

- Despite trends towards worse survival with higher Snord67 expression, no statistically significant associations were observed in the other subtypes. However, we also observed that high expression of both Snord67 and its host gene *CKAP5* was highly associated with significantly decreased overall survival in the HER2-enriched subtype of breast cancer (panel c, new Supplemental Figure 12G in the manuscript). The relevance of combined *CKAP5* and Snord67 expression is shown by a highly significant correlation between the two (Point 2 *below* and new Supplemental Figure 12F in the manuscript). Although we have shown in several models that loss of Snord67 impacted phenotypes independent of *CKAP5* host gene expression, the correlation of Snord67 and *CKAP5* expression in breast cancer patients raises the possibility that Snord67 expression may be co-regulated with expression of its host gene. These results suggest that high Snord67 expression is a prognostic marker in some subtypes of breast cancer. Therefore, Snord67 expression appears to be relevant in human patients.

a. Relative Snord67 expression by breast cancer subtype, *FDR-adjusted p -value <0.05 . b. Overall survival stratified by Snord67 expression in the luminal B subtype. c. Overall survival stratified by Snord67 expression in the HER2-enriched subtype.

(2) SNORD67 expression is both up- and down-regulated across 28 patients suggests it may not be critical to LN mets. While it might be relevant to some patients, a more systematic analysis in patients would have been able to demonstrate whether other RNAs have a more robust and consistent influence on LN mets or if SNORD67 is the optimal candidate.

Although Snord67 expression may not be relevant for every patient with breast cancer, our analysis of the TCGA BRCA SNORic data suggests that Snord67 expression may have prognostic significance in patients with aggressive subtypes of breast cancer (see response to Comment #1 above). There is significant heterogeneity in breast cancer, and multiple mechanisms contribute to breast cancer lymphatic metastasis. This heterogeneity is also evidenced by our murine models, where Snord67 was upregulated in only a subset of *de novo* lymph node subclones relative to mammary fat pad derived subclones, despite being derived from an isogenic cell line (Figure 1D). Notably, in breast cancer patients, Snord67 expression significantly correlates with expression of its host gene *CKAP5* (see figure panel, new Supplemental Figure 12F in the manuscript). This correlation suggests a possible mechanism by which Snord67 may be upregulated in the lymph node metastases in some patients but not others. However, the precise mechanisms underlying the regulation of Snord67 in breast cancer lymph node metastases remain unclear and are outside the scope of our manuscript.

Correlation between *CKAP5* expression and Snord67 expression in breast cancer patients.

(3) The authors have added a correlation between SNORD67 and AS. However, given that SNORD67 can both be up- and down-regulated, it will be important to establish what levels of SNORD67 alter splicing sufficiently to promote LN mets.

To determine the level of Snord67 that is required to promote the observed changes in alternative splicing, we determined the optimal Snord67 expression threshold to detect a significant change in the PSI of *MYO18A* exon 40 and *NFYA* exon 3 by t-test. We found that a Snord67 expression threshold of $-\Delta\text{Ct}(\text{Snord67}/\text{Gapdh}) = -1.83$ was optimal to detect a change in PSI of *MYO18A* exon 40 in the low Snord67 and high Snord67 expression groups (panel a, new Supplemental Figure 8C in the manuscript). A Snord67 expression threshold of $-\Delta\text{Ct}(\text{Snord67}/\text{Gapdh}) = -1.465$ was optimal to detect a change in PSI of *NFYA* exon 3 in the low Snord67 and high Snord67 expression groups (panel b, new Supplemental Figure 8C in the manuscript). This analysis estimates the level of Snord67 expression that is associated with specific changes in alternative splicing in patients with breast cancer. These changes in alternative splicing may in turn contribute to lymphatic metastasis.

a. Pearson correlation of *MYO18A* exon 40 PSI with Snord67 expression (*left*). The difference in *MYO18A* exon 40 PSI in low vs. high Snord67 expression groups is significant by t-test (*right*). **b.** Pearson correlation (*left*) and t-test (*right*) for the association of *NFYA* exon 3 PSI with Snord67 expression.

Reviewer #2 (Remarks to the Author):

The authors have made substantial improvements to the manuscript, particularly in convincingly demonstrating that the expression of Sno host genes is unaffected by the knockouts and in enhancing the clarity of the figure descriptions.

Thank you for your supportive comments.

However, the revision introduces two new figures (Fig 7 and 8) that still require refinement. In these figures, the authors have successfully inactivated Snord67 in human MDA-MB-231-D3H2LN-luc WT cells (D3H2), which represents a valuable new reagent. Despite this, the examples presented in Figures 8A and 8B are not entirely convincing. The observed changes in PSI values are minimal, and the effects are not clearly reflected in the validation PCRs. This may be due to the authors' focus on identifying events that occur both in the cell lines and tumors, and it is probably possible to find better examples without that constrain.

Rather than concentrating on specific alternative splicing events, the authors might consider exploring the potential for reduced splicing efficiency or precision following Snord67 inactivation (or reduced expression). A modified version of Figure 7C, illustrating the number of skipped exons and retained introns under both conditions, could effectively demonstrate reduced splicing activity (as exon skipping and intron retention require fewer splicing events). Additionally, quantifying the total number of split reads and non-annotated splicing events could provide insights into splicing efficiency and precision, respectively (this analysis can be performed using RegTools). Furthermore, calculating the "strength" of de novo splice sites could help identify any reduction in splicing precision.

The goal of these bioinformatics approaches would be to determine whether reduced splicing efficiency, rather than specific splicing events, is observed consistently both in vitro and in tumor samples.

Thank you for the suggestion to investigate splicing efficiency. To explore an effect of Snord67 on global splicing efficiency, we examined the intron retention ratio in wild-type and Snord67 knockout cells. This was done for all introns. Interestingly, we found that intron retention ratios were significantly lower in Snord67 knockout cells. The cumulative distribution of intron retention ratios was significantly different in Snord67 knockout cells compared to 4T1 WT cells by the Kolmogorov–Smirnov test (panel a, new Figure 6G in the manuscript). Another way of visualizing this data is to plot the splicing efficiency (defined as $1 - \text{intron retention ratio}$) of Snord67 knockout cells compared to 4T1 WT cells (panels b–c, new Supplemental Figure 10D in the manuscript), demonstrating higher splicing efficiency (= less intron retention) in Snord67 knockout cells compared to 4T1 WT. These results suggest that Snord67 promotes intron retention and reduces splicing efficiency. While splicing efficiency would ideally be determined using methodologies that involve sequencing nascent RNA, we feel that such studies are beyond the scope of this study.

a. Cumulative distribution of intron retention ratio in 4T1 WT, Snord67KO-1, and Snord67KO-2 cells. **b.** Splicing efficiency in Snord67KO-1 vs. 4T1 WT cells. **c.** Splicing efficiency in Snord67KO-2 vs. 4T1 WT cells.

Reviewer #3 (Remarks to the Author):

My concerns have been addressed. Thank you!

Thank you. Your constructive comments led us to perform additional experiments that significantly improved the quality of our manuscript.

We appreciate the constructive suggestions of the reviewers for extending the scope and detail of these studies. Thank you in advance for your thoughtful consideration. Please let me know if I can provide any further information.

Dear Colleagues:

We thank the reviewers for their comments. We have addressed the concerns raised by the reviewers. Point-by-point responses are in blue below.

Reviewer #1 (Remarks to the Author):

The authors have sufficiently addressed my concerns.

Thank you. Your constructive comments led us to perform additional experiments that significantly improved the quality of our manuscript.

Reviewer #2 (Remarks to the Author):

The claim that Snord67 specifically affects alternative splicing, rather than splicing in general, still does not appear to be strongly supported by the data. Previous comments emphasized the need to examine D3H2 cells inactivated for Snord67 to assess its impact on splicing efficiency. Instead, the authors have focused on 4T1 cells and now provide new data showing a clear overall impact of Snord67 on splicing efficiency. Yet, the conclusions remain centered on "alternative splicing," despite the relatively weak effects observed. In this regard, it should be noted that, as in the previous version, the examples of alternative splicing in Figures 8A and 8B show only minimal changes in PSI values, and these effects are not clearly reflected in the validation PCRs. This issue remains to be addressed.

Main point: A revision of the conclusions, rebalancing the emphasis on splicing versus alternative splicing, is strongly recommended before publication.

We have revised the text in the Results and Discussion sections to rebalance the emphasis on splicing efficiency versus alternative splicing. Specifically, we revised our language to more broadly refer to changes in the splicing landscape rather than changes in alternative splicing specifically, and we shortened the discussion of alternative splicing of *MYO18A* exon 40 and *NFYA* exon 3. Finally, we expanded the discussion of the potential contribution of changes in splicing efficiency to Snord67-dependent phenotypes (pp. 21–22):

“We found that loss of Snord67 led to overall increased splicing efficiency, as evidenced by decreased intron retention. Intron retention is frequently increased in cancer compared to normal tissues and is a common mechanism for tumor suppressor inactivation in cancer cells. In patients with cancer, somatic single-nucleotide variants that cause intron retention are enriched in tumor suppressor genes, where they frequently lead to introduction of a premature termination codon and result in tumor suppressor inactivation. In addition to genetic alterations, protein binding and post-transcriptional modification can target tumor suppressor transcripts for intron retention and nuclear decay, thereby promoting cancer growth. Here, we suggest that Snord67-guided 2'-O-methylation of a core spliceosomal component, U6 snRNA, facilitates intron retention. While the downstream effects on tumor suppressor gene expression remain unknown, this global effect on intron retention may contribute to cancer cell survival and egress from lymph nodes, either through tumor suppressor inactivation or through other mechanisms.”